# Entanglement wedge in flat holography and entanglement negativity

**Debarshi Basu, Ashish Chandra, Vinayak Raj and Gautam Sengupta⋆**

Department of Physics, Indian Institute of Technology, Kanpur 208 016, India

⋆ sengupta@iitk.ac.in

## Abstract

We establish a construction for the entanglement wedge in asymptotically flat bulk geometries for subsystems in dual $(1 + 1)$-dimensional Galilean conformal field theories in the context of flat space holography. In this connection we propose a definition for the bulk entanglement wedge cross section for bipartite states of such dual non relativistic conformal field theories. Utilizing our construction for the entanglement wedge cross section we compute the entanglement negativity for such bipartite states through the generalization of an earlier proposal, in the context of the usual $AdS/CFT$ scenario, to flat space holography. The entanglement negativity obtained from our construction exactly reproduces earlier holographic results and match with the corresponding field theory replica technique results in the large central charge limit.

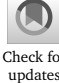
# 1  Introduction

Quantum entanglement in extended many body systems has emerged as an exciting issue attracting intense research attention in the recent past across a diverse variety of disciplines. We know from quantum information theory that the entanglement of bipartite pure states is characterized by the entanglement entropy given by the von Neumann entropy of the reduced density matrix. However the description of mixed state entanglement is an involved issue as the entanglement entropy receives contributions from irrelevant correlations and is not a suitable entanglement measure for such mixed states. Several alternate measures to describe mixed state entanglement has been proposed in quantum information theory but are usually difficult to compute as they involve optimization over LOCC protocols. In this context Vidal and Warner in their seminal work [1] proposed a novel computable measure for mixed state entanglement termed *entanglement negativity* which provided an upper bound on the distillable entanglement for a bipartite mixed state. Entanglement negativity was defined as the logarithm of the trace norm of the reduced density matrix, partially transposed with respect to one of the subsystems. Remarkably, in a series of communications [2–4], the authors computed the entanglement negativity for various bipartite states in $(1 + 1)$-dimensional conformal field theories ($CFT_{1+1}$) through a replica technique similar to an earlier technique for the entanglement entropy described in [5,6].

In the framework of the $AdS/CFT$ correspondence, Ryu and Takayanagi [7, 8] (RT) proposed a holographic conjecture to compute the universal part of the entanglement entropy of a subsystem in $CFT_d$s dual to bulk static $AdS_{d+1}$ geometries which was proportional to the area of a bulk codimension-2 minimal surface (RT surface) homologous to the subsystem. A generalization of this RT conjecture for $CFT_d$s dual to non-static bulk $AdS_{d+1}$ geometries was proposed by Hubeny, Rangamani and Takayanagi (HRT) in [9]. Both the RT and HRT conjectures were subsequently proved in a series of significant works described in [10–16]. Following an earlier attempt described in [17], holographic proposals for the entanglement negativity of various bipartite states in dual $CFT_d$s were proposed in a series of interesting works [18–29] where the authors utilized algebraic sums of the areas of the (H)RT surfaces homologous to certain combinations of subsystems relevant to the mixed states in question. A substantiation for the holographic prescription in the context of the $AdS_3/CFT_2$ scenario described in [18] was provided in [30] through a large central charge analysis of the entanglement negativity for dual $CFT_{1+1}$s using the monodromy technique developed in [13, 31, 32] which may be also suitably extended to other subsystem configurations. For the corresponding higher dimensional $AdS_{d+1}/CFT_d$ scenario a proof restricted to spherical entangling surfaces may be inferred from the arguments recently described in [33] based on certain novel replica symmetry breaking saddles for the bulk gravitational path integral [34]. However the corresponding arguments for general subsystem geometries is still a non trivial open issue.

It should be noted here that the authors in [35] have advanced an alternative approach to compute the holographic entanglement negativity for bipartite states in the $AdS_{d+1}/CFT_d$ scenario, involving the backreacted bulk entanglement wedge cross section (EWCS). Subsequently in another recent communication [36] a *proof* for this proposal based on the reflected entropy [37] was advanced. Interestingly these two distinct proposals for the holographic entanglement negativity were equivalent upto certain overall multiplicative factors arising from the backreaction as discussed in [38]. A covariant version of the proposal based on the entanglement wedge described above, was established recently in [39] where the authors obtained the holographic entanglement negativity for bipartite states in $CFT_{1+1}$s dual to non-static bulk $AdS_3$ geometries through the extremal EWCS. Furthermore the entanglement wedge was proposed to be the bulk dual for the reduced density matrix of the corresponding $CFT$ subsystem [40–43]. In [44,45] the EWCS was conjectured to be equal to the holographic entanglement of purification (EoP) which, in contrast to the entanglement negativity, receives contributions from both quantum and classical correlations [46]. The EWCS has also been proposed to be holographically related to other entanglement measures such as the odd entanglement entropy [47], the balanced partial entanglement entropy [48] and the reflected entropy [37,49,50] for dual $CFT_d$s. This naturally makes the EWCS an interesting bulk quantity for investigation in holographic quantum entanglement.

On a separate note non-relativistic version of (1+1)-dimensional field theories with Galilean conformal symmetry were obtained in [51–54] through the İnönü-Wigner contraction of the relativistic conformal algebra for $CFT_{1+1}$s. This involved different scalings of the space and the time coordinates to arrive at the corresponding non relativistic limit. A suitable replica technique was developed to compute the entanglement entropy of bipartite states in such Galilean conformal field theories ($GCFT_{1+1}$) in [54,55]. Subsequently another replica technique was established in [56] to compute the entanglement negativity for bipartite states in such $GCFT_{1+1}$s. In the framework of flat space holography [57,58], it was proposed that the above mentioned $GCFT_{1+1}$s are dual to gravitational theories in (2 + 1)-dimensional bulk asymptotically flat spacetime [59]. The Galilean conformal algebra in (1+1)-dimensions ($GCA_{1+1}$) was isomorphic to the infinite-dimensional Bondi-Metzner-Sachs ($BMS_3$) algebra which described the asymptotic symmetry of (2 + 1)-dimensional bulk flat geometries. The holographic entanglement entropy for a single interval in the $BMS_3$ field theory located at the

null infinity of the bulk asymptotically flat geometry was computed in [60]. Remarkably, similar results were reproduced[1] in [61] using a flat holographic version of the HRT prescription first proposed in [60]. Their construction utilized null geodesics dropping from the (properly regulated) end points of the interval under consideration living at the asymptotic boundary of the bulk spacetime, which were connected inside the bulk through an extremal curve. These null geodesics were shown to be tangential to the bulk modular flow vector. The fixed points of the modular flow vector constitute the connecting extremal curve which compute the holographic entanglement entropy [60]. Subsequently in [62], a suitable bulk prescription to compute the generalized gravitational entropy for generic spacetimes with non-Lorentz invariant dual field theories was advanced which extended the causal structure of the boundary theory into the bulk. Furthermore several subtleties involved in this construction were investigated in [63] with subsequent generalization to higher dimensions.

As described earlier the entanglement entropy was not a viable measure for mixed entanglement. In this context, given the developments described above, a holographic description of mixed state entanglement in such $GCFT_{1+1}$s dual to asymptotically flat bulk geometries was a significant open issue. This interesting issue was investigated recently in [64] where an elegant and clear holographic characterization for the entanglement negativity of bipartite states in a class of $GCFT_{1+1}$s dual to bulk asymptotically flat (2+1)-dimensional Einstein gravity and topologically massive gravity (TMG) was established. Their construction involved specific algebraic sums of the lengths of extremal curves (HRT like surfaces termed *swing surfaces* [65]) homologous to certain combinations of intervals appropriate to the bipartite state in question. Interestingly the holographic entanglement negativity for such bipartite states in dual $GCFT_{1+1}$s exactly reproduced the corresponding replica technique results [56] in the large central charge limit. Their results were further substantiated through a rigorous large central charge analysis utilizing the geometric monodromy technique developed in [66]. Recently the above mentioned holographic proposals have been reformulated in the context of asymptotically flat generalized minimal massive gravity in [67] where the authors have reproduced the corresponding field theory results for the entanglement negativity [56]. This serves as yet another consistency check for the above holographic proposals and also advances our understanding of the flat space holography.

As mentioned earlier in the usual $AdS/CFT$ framework the holographic entanglement negativity for bipartite states in dual $CFT_{1+1}$s could be characterized through the bulk EWCS as described in [35, 36]. Naturally the extension of this construction to flat holography is a significant issue which needs to be investigated. In this article, we address this outstanding issue and establish a novel construction for the bulk entanglement wedge and provide a definition for the EWCS in bulk asymptotically flat (2+1)-dimensional Einstein gravity and TMG dual to $GCFT_{1+1}$s. Subsequently we compute the holographic entanglement negativity for various bipartite states in such $GCFT_{1+1}$s through the EWCS generalizing the construction of [35, 36] to the framework of flat holography. Remarkably the holographic entanglement negativity computed through our construction matches up to an additive constant with the results described in [64] from the alternative proposal involving the algebraic sum of the lengths of specific combinations of bulk extremal curves as well as with the field theory replica technique results described in [56] in the large central charge limit. This provides an extremely strong substantiation and consistency check for our construction of the bulk EWCS and the consequent computation for the entanglement negativity in the context of flat holography. Recently in the context of $AdS/CFT$ framework, an interpretation for this additive constant was provided in [68] in terms of the fidelity of a particular Markov recovery process which also has a consistent bulk description in terms of the number of (non-trivial) boundaries of the EWCS. Interestingly, our findings conform to the above geometric interpretation which provides further

---

[1]Interestingly, the consistency of such calculations may be checked against the Wilson line computations in [55].

support to our holographic construction. Note that this quantity was also examined in [48] in the context of the canonical purification of a mixed state $\rho_{AB}$ by introducing candidate purifiers $A'$ and $B'$. The author furthermore defined and investigated a new quantum information theoretic measure called the balanced partial entanglement (BPE) and argued for a putative duality with the EWCS. To this end, the above mentioned difference has an alternative interpretation in terms of the partial entanglement entropy (PEE) between $A$ and $B'$ or equivalently $B$ and $A'$, and was termed as the crossing PEE in [48].

This article is organized as follows. In section 2 we present a brief review of holographic entanglement in flat geometries and collect certain results required for our purpose. Subsequently in section 3 we propose a generalization of the relation between the EWCS and the holographic entanglement negativity [35, 36], to flat space holography. In section 4 we describe our construction for the bulk EWCS in flat Einstein gravity dual to a $GCFT_{1+1}$ and compute the holographic entanglement negativity for various bipartite states. Furthermore in section 5 we extend our construction for the EWCS to bulk topologically massive gravities (TMGs) and subsequently compute the holographic entanglement negativity for various bipartite states in the corresponding dual $GCFT_{1+1}$s. We then summarize our results and present our discussions in section 6. Additionally in Appendix A, we demonstrate that the bulk EWCS for the asymptotically flat spacetimes may also be obtained through a limiting analysis of the corresponding $AdS_3/CFT_2$ results. Finally in Appendix B we obtain the entanglement negativity for bipartite states in $GCFT_{1+1}$s dual to the special case of bulk flat space TMG known as flat chiral gravity.

## 2   Review of Holographic Entanglement in flat geometries

In this section we briefly review the (1+1)-dimensional Galilean conformal field theory ($GCFT_{1+1}$). Subsequently, we review the computation of extremal curves homologous to an interval at the asymptotic boundary in flat space holography. We finish this section with a very short review of the entanglement negativity as an entanglement measure for mixed state configurations.

### 2.1   Galilean Conformal Field Theory

In this subsection we will briefly review the silent features of the $GCFT_{1+1}$. In this context, the Galilean conformal algebra ($GCA_2$) in two dimensions may be obtained from the relativistic conformal algebra through İnönü-Wigner contraction, this requires the rescaling of the space and the time coordinates as

$$t \to t, \qquad x \to \epsilon x, \tag{1}$$

with $\epsilon \to 0$. This is equivalent to the small velocity limit $v \sim \epsilon$. In the plane representation, the generators of the $GCA_2$ are given as [55]

$$L_n = t^{n+1}\partial_t + (n+1)t^n x \partial_x, \quad M_n = t^{n+1}\partial_x. \tag{2}$$

These generators lead to the following Lie algebra with a central extension

$$[L_n, L_m] = (m-n)L_{n+m} + \frac{c_L}{12}(n^3-n)\delta_{n+m,0},$$

$$[L_n, M_m] = (m-n)M_{n+m} + \frac{c_M}{12}(n^3-n)\delta_{n+m,0}, \tag{3}$$

$$[M_n, M_m] = 0,$$

where $c_L$ and $c_M$ are the central charges for the $GCA_2$. In this work, we will also use the cylinder representation of the algebra given by the generators [51]

$$L_n = e^{in\phi}\left(\partial_\phi + inu\partial_u\right), \quad M_n = e^{in\phi}\partial_u. \tag{4}$$

The plane and cylinder representations are related by the following transformation [69]

$$x = e^{i\phi}, \quad t = iu\,e^{i\phi}. \tag{5}$$

The highest weight representation of the $GCA_2$ are labelled by the eigenvalues $\Delta$ and $\chi$ of the maximally commuting generators $(L_0, M_0)$ [53] as

$$L_0\,|\Delta,\chi\rangle = \Delta\,|\Delta,\chi\rangle, \quad M_0\,|\Delta,\chi\rangle = \chi\,|\Delta,\chi\rangle. \tag{6}$$

For the primary fields $V(x,t)$, the four point function in the $GCFT_{1+1}$ may be expressed as [52]

$$\left\langle\prod_{i=1}^{4} V_i(x_i, t_i)\right\rangle = \prod_{1\le i < j\le 4} t_{ij}^{\frac{1}{3}\sum_{k=1}^4 \Delta_k - \Delta_i - \Delta_j} e^{-\frac{x_{ij}}{t_{ij}}\left(\frac{1}{3}\sum_{k=1}^4 \chi_k - \chi_i - \chi_j\right)} \mathcal{G}_{GCA}\left(T, \frac{X}{T}\right), \tag{7}$$

where $(\Delta_i, \chi_i)$ are the conformal weights of the primary fields $V_i(x_i, t_i)$ with $(i = 1, 2, 3, 4)$ and $\mathcal{G}_{GCA}$ is the non-universal function of cross-ratio which depends on the full operator content of the specific field theory. In eq. (7), $T$ and $\frac{X}{T}$ are the non-relativistic cross-ratios that are given as

$$T = \frac{t_{12}t_{34}}{t_{13}t_{24}}, \quad \frac{X}{T} = \frac{x_{12}}{t_{12}} + \frac{x_{34}}{t_{34}} - \frac{x_{13}}{t_{13}} - \frac{x_{24}}{t_{24}}. \tag{8}$$

In the large central charge limit, $\mathcal{G}_{GCA}$ has the following expression [52,61]

$$\mathcal{G}_{GCA}\left(T, \frac{X}{T}\right) \approx \sum_p (1-T)^{\Delta_p}\left(1+\sqrt{T}\right)^{-2\Delta_p} e^{\chi_p \frac{X}{\sqrt{T}(1-T)}}, \tag{9}$$

where the sum is over the Galilean conformal blocks corresponding to the complete set of intermediate states. As described in [69], the $GCFT_{1+1}$ primaries transform from the plane to the cylinder representation as

$$V'(\phi, u) = A\,e^{i\phi\Delta}\,e^{iu\chi}\,V(x, t), \tag{10}$$

where $A$ is a phase factor. Using eq. (10), one can deduce the correlation functions on the cylinder. In the cylinder representation, the cross-ratios eq. (8) becomes

$$T = \frac{\sin\frac{\phi_{12}}{2}\sin\frac{\phi_{34}}{2}}{\sin\frac{\phi_{13}}{2}\sin\frac{\phi_{24}}{2}}, \quad \frac{X}{T} = \frac{1}{2}\left(\frac{u_{12}}{\tan\frac{\phi_{12}}{2}} + \frac{u_{34}}{\tan\frac{\phi_{34}}{2}} - \frac{u_{13}}{\tan\frac{\phi_{13}}{2}} - \frac{u_{24}}{\tan\frac{\phi_{24}}{2}}\right). \tag{11}$$

We will use eqs. (8) and (11) for the cross-ratios in respective representations to compute the holographic entanglement negativity for various subsystem configurations in the dual $GCFT_{1+1}$s.

## 2.2 Extremal curves in flat geometries

We now give a brief review of the construction for the extremal curves homologous to an interval in a holographic $GCFT_{1+1}$ in the context of asymptotically flat gravity. To this end we consider an interval $A$ in a $(1+1)$-dimensional $GCFT$ dual to the $(2+1)$-dimensional flat Einstein gravity for which the line element in the Eddington-Finkelstein coordinates reads

$$ds^2 = -du^2 - 2\,du\,dr + r^2\,d\phi^2, \tag{12}$$

with the retarded time $u = t - r$. The dual $GCFT_{1+1}$ is at the future null infinity $r \to \infty$ with $u$ and $\phi$ fixed. The asymptotic symmetry analysis at null infinity of flat Einstein gravity gives the Galilean conformal algebra with only one non-zero central charge given as [54, 70, 71]

$$c_L = 0, \quad c_M = \frac{3}{G_N}. \tag{13}$$

The boosted interval $A = [(u_1, \phi_1), (u_2, \phi_2)]$ is at the future infinity with endpoints denoted by $\partial_i A$ ($i = 1, 2$). Since this interval is not at a constant time slice, we use a flat space generalization of the HRT construction [9] which describes a bulk codimension-2 extremal surface homologous to the boundary subsystem for non static bulk geometries [60, 61].

The codimension-2 extremal surface in the (2+1)-dimensional asymptotically flat geometry becomes an extremal curve. The bulk extremal curve in this context of flat space holography is segmented and consists of two null geodesics labelled as $\gamma_i$ descending from the endpoints $\partial_i A$ of the interval $A$. However these null geodesics do not intersect in general, as shown in figure 1. A third bulk curve is required to connect them which is uniquely specified by constraining the distance between the two null geodesics $\gamma_i$s to be extremal. Note that the extremal curve between any two points in the above construction is a straight line as discussed in [61]. The extremal length of the curve segments homologous to the interval $A$ at the null infinity is just given by the length of the extremal curve connecting the points $y_i \in \gamma_i$ which can be evaluated to be [61]

$$L_{\text{total}}^{\text{extr}} = L^{\text{extr}}(y_1, y_2) = \left| \frac{u_{21}}{\tan \frac{\phi_{12}}{2}} \right|, \tag{14}$$

where $u_{ij} = u_i - u_j$ and $\phi_{ij} = \phi_i - \phi_j$. Note that the holographic entanglement entropy may be computed using this extremal length as [60, 61]

$$S_A = \frac{1}{4G_N} L_{\text{total}}^{\text{extr}}. \tag{15}$$

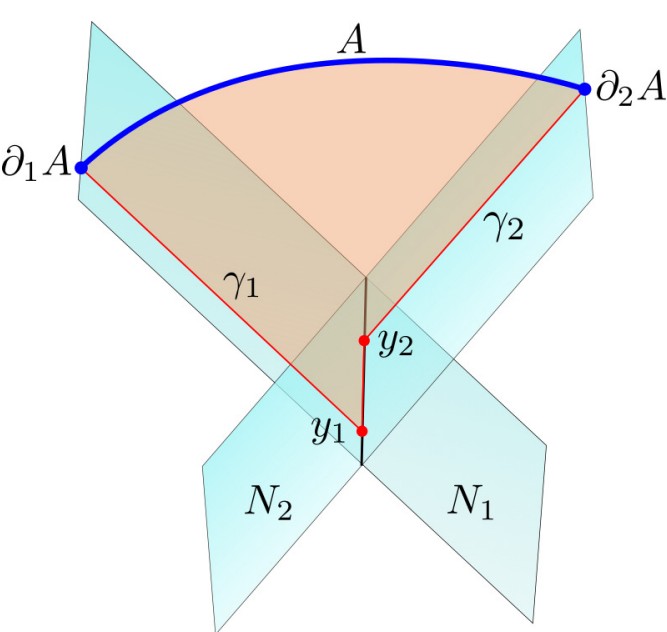

Figure 1: Extremal curve homologous to an interval in a $GCFT_{1+1}$ dual to Einstein gravity in asymptotically flat spacetime. Figure modified from [61].

Remarkably a bulk proof of the above geometric picture for the holographic entanglement entropy involving gravitational path integrals was proposed in [62], following the Lewkowycz-Maldacena procedure [14] in the context of flat space holography. For an interval at the asymptotic boundary of the spacetime, their prescription involves a corresponding bulk extremal curve located at the intersection of two associated null hypersurfaces which extends the boundary causal structure to the bulk. In this geometric picture, the null geodesics located on such null hypersurfaces are the lines emanating from the boundary endpoints and moving along the bulk modular flow vector. The segment of the extremal curve connecting the null geodesics corresponds to the fixed points of the bulk modular flow vector. This construction provides a natural justification for the null segments for the extremal curve homologous to the interval $A$ described earlier.

In the following, we will only be interested in the length of the extremal curve homologous to an interval at the null infinity of the bulk flat gravitational theory and omit the subsequent discussion of entanglement entropy.

We can similarly compute the length of the extremal curve homologous to a boosted interval in a finite temperature $GCFT_{1+1}$ dual to the bulk non-rotating flat space cosmologies (FSC) in (2+1)-dimensions whose metric is given by [54, 55, 69]

$$ds^2 = M \, du^2 - 2 \, du \, dr + r^2 d\phi^2 \,, \tag{16}$$

where $M$ is the ADM mass of the spacetime and is related to the temperature in the dual $GCFT_{1+1}$ at null infinity as

$$M = \left(\frac{2\pi}{\beta}\right)^2 . \tag{17}$$

The length of the extremal curve homologous to an interval $A$ in the thermal $GCFT_{1+1}$ may then be computed to be [61]

$$L_{\text{total}}^{\text{extr}} = \frac{2\pi u_{12}}{\beta} \coth\left(\frac{\pi \phi_{12}}{\beta}\right) . \tag{18}$$

Finite sized version of this computation, as performed in [64], involves a $GCFT_{1+1}$ compactified on a spatial circle of circumference $L_\phi$. The bulk dual geometry is the global Minkowski orbifold, which is a quotient of the usual Minkowski spacetime with a compact spatial circle $(u, \phi) \sim (u, \phi + L_\phi)$, with the metric [60, 61]

$$ds^2 = -\left(\frac{2\pi}{L_\phi}\right)^2 du^2 - 2du \, dr + r^2 d\phi^2 \,, \tag{19}$$

where again the size of the subsystem $L_\phi$ is related to the ADM mass of the spacetime $M$ as

$$M = -\left(\frac{2\pi}{L_\phi}\right)^2 . \tag{20}$$

The length of the extremal curve homologous to an interval $A$ in the dual $GCFT_{1+1}$ may then be obtained following a similar procedure as the previous cases to be [64]

$$L_{\text{total}}^{\text{extr}} = \frac{2\pi u_{12}}{L_\phi} \cot\left(\frac{\pi \phi_{12}}{L_\phi}\right) . \tag{21}$$

This establishes the construction of the HRT-like extremal curves [61, 64] homologous to an interval in $GCFT_{1+1}$s dual to (2+1)-dimensional asymptotically flat bulk theories described by Einstein gravity. We will use some of the results mentioned above in our computation of the EWCS in section 4.

## 2.3 Entanglement Negativity

As mentioned earlier, in quantum information theory the entanglement entropy is not a valid measure for the entanglement of mixed state configurations. However the authors in [1] proposed a computable measure for mixed state entanglement known as entanglement negativity. This entanglement measure is defined as the trace norm of the partial transpose of the density matrix with respect to one of the subsystems. In this context, we start with a tripartite system in a pure state which involves the subsystems $A_1$, $A_2$ and $B$. Subsequently we trace over the subsystem $B$ to find the reduced density matrix of the mixed state configuration ($A = A_1 \cup A_2$) described as $\rho_A = \text{Tr}_B \rho$, where $\rho$ is the density matrix of tripartite state $A \cup B$. Therefore the entanglement negativity of the bipartite mixed state is defined as

$$\mathcal{E} = \ln \text{Tr} ||\rho_A^{T_2}||, \tag{22}$$

where the trace norm $\text{Tr} ||\rho_A^{T_2}||$ is defined as the sum of absolute eigenvalues of $\rho_A^{T_2}$. The partial transpose of the reduced density matrix $\rho_A$ is described as

$$\left\langle e_i^{(1)} e_j^{(2)} \left| \rho_A^{T_2} \right| e_k^{(1)} e_l^{(2)} \right\rangle = \left\langle e_i^{(1)} e_l^{(2)} \left| \rho_A \right| e_k^{(1)} e_j^{(2)} \right\rangle, \tag{23}$$

where the basis vectors $\left| e_i^{(1)} \right\rangle$ and $\left| e_j^{(2)} \right\rangle$ are the elements of the Hilbert spaces $\mathcal{H}_1$ and $\mathcal{H}_2$ corresponding to the subsystems $A_1$, $A_2$.

Recently the authors of [56] constructed an appropriate replica technique to compute the entanglement negativity in $GCFT_{1+1}$, following the corresponding works [2–4] in the relativistic scenario. Their analysis essentially involves a replica manifold $\Sigma_{n_e}$ consisting of $n_e$ copies ($n_e$ even) of the $GCFT_{1+1}$ plane sewed together along branch cuts present on the subsystems under consideration, in a particular orientation [3, 56]. On $\Sigma_{n_e}$ the replica partition function $\text{Tr}(\rho_A^{T_2})^{n_e}$ is given by a four point function of twist fields $\Phi_{\pm n_e}$ inserted at the endpoints of the intervals $A_1 = [(x_1, t_1), (x_2, t_2)]$ and $A_2 = [(x_3, t_3), (x_4, t_4)]$:

$$\text{Tr}(\rho_A^{T_2})^{n_e} = \left\langle \Phi_{n_e}(x_1, t_1) \Phi_{-n_e}(x_2, t_2) \Phi_{-n_e}(x_3, t_3) \Phi_{n_e}(x_4, t_4) \right\rangle. \tag{24}$$

Finally, the entanglement negativity between the two intervals $A_1$ and $A_2$ was computed through the replica limit

$$\mathcal{E} = \lim_{n_e \to 1} \ln \text{Tr}(\rho_A^{T_2})^{n_e}. \tag{25}$$

Subsequently in [64] the authors proposed several holographic constructions for the entanglement negativity in the asymptotically flat spacetime for different bipartite pure and mixed states. In particular their proposals involve specific linear combinations of the lengths of the extremal curves homologous to various subsystems under consideration[2]. This is reminiscent of the earlier work on holographic entanglement negativity in the relativistic setup [19–21]. In this work, we propose an alternative construction to compute the holographic entanglement negativity through an entanglement wedge cross-section in the asymptotically flat spacetimes based on similar reasoning in the relativistic scenario [35, 36].

## 3 Connection between the EWCS and the entanglement negativity in flat geometry

In the proceeding subsection we briefly review the evaluation of the bulk entanglement wedge cross section [37, 44, 45, 47, 49, 50, 72, 73] for the $AdS_{d+1}/CFT_d$ scenarios which has been

---

[2]Such linear combinations of the extremal curve lengths may be re-expressed in terms of the holographic mutual information between various subsystems utilizing the flat holographic version of the HRT prescription in eq. (15) [56, 64].

proposed to be dual to the entanglement of purification (EoP) [44,45]. The authors in [35,36] proposed that the holographic entanglement negativity for subsystems in $CFT_d$s dual to bulk static $AdS_{d+1}$ geometries could be obtained through the EWCS. Subsequently, we propose an extension of this connection between the holographic entanglement negativity and the EWCS in the context of flat space holography.

## 3.1 Entanglement Wedge Cross Section in $AdS_{d+1}/CFT_d$ framework

We begin with a short review of the entanglement wedge and the entanglement wedge cross section in the framework of $AdS_{d+1}/CFT_d$. To this end we consider the bipartite mixed state of two disjoint subsystems $A$ and $B$ at a constant time slice in a $CFT_d$ dual to the bulk static $AdS_{d+1}$ geometry. Let the RT surface for the subsystem $A \cup B$ be labelled as $\Gamma_{AB}$. Then the entanglement wedge $M_{AB}$ for the subsystem $A \cup B$ on this constant time slice is defined to be the codimension-1 spatial bulk region bounded by $A \cup B \cup \Gamma_{AB}$. The field theory dual of the entanglement wedge was shown to be the corresponding reduced density matrix $\rho_{AB}$ [40–42]. Now the minimal entanglement wedge cross section (EWCS) is described by the area of the bulk codimension-2 minimal surface $\Sigma_{AB}^{\min}$ which bisects the entanglement wedge $M_{AB}$ in two parts $A \cup \Gamma_{AB}^{(A)}$ and $B \cup \Gamma_{AB}^{(B)}$ containing the subsystems $A$ and $B$ respectively as shown in figure 2. The entanglement wedge cross section $E_W$ for the subsystem $A \cup B$ is then defined as follows

$$E_W(\rho_{AB}) = \min_{\Gamma_{AB}^{(A)} \subset \Gamma_{AB}^{\min}} \left[ \frac{\text{Area}(\Sigma_{AB})}{4G_N} \right], \tag{26}$$

where $G_N$ is the Newton's constant. The EWCS have been conjectured to be the bulk dual to the *entanglement of purification* (EoP) as it satisfies identical properties [44,45].

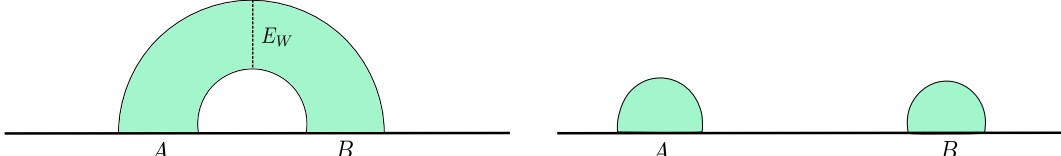

Figure 2: Left: The green shaded region represents the entanglement wedge for the subsystem $A \cup B$. The dotted line gives the EWCS. Right: If the subsystems $A$ and $B$ are sufficiently far away, their entanglement wedge remains disconnected and the EWCS vanishes. (Adapted from [39])

It should be noted here that in [35,36], it was proposed that one could obtain the holographic entanglement negativity for subsystems in $CFT_d$s dual to bulk static $AdS_{d+1}$ geometries, from the area of the backreacted minimal EWCS. For a special class of subsystems having a spherical entangling surface in the $CFT_d$s for which the effect of backreaction is known, the holographic entanglement negativity is given in terms of the backreacted EWCS as follows [35,36]

$$\mathcal{E}(A:B) = \mathcal{X}_d \, E_W(\rho_{AB}), \tag{27}$$

where $\mathcal{X}_d$ is a dimension dependent constant which accounts for the backreaction of the bulk cosmic brane for the conical defect in the replica limit ($n \to 1$). The expression for $\mathcal{X}_d$ for a pure vacuum state in a $CFT_d$ was determined to be [17,74]

$$\mathcal{X}_d = \left( \frac{1}{2} x_d^{d-2}(1 + x_d^2) - 1 \right), \quad x_d = \frac{2}{d} \left( 1 + \sqrt{1 - \frac{d}{2} + \frac{d^2}{4}} \right). \tag{28}$$

The $AdS_3/CFT_2$ framework is a special case where the entangling surfaces of all subsystems in the dual field theory have spherical geometries. Therefore the holographic entanglement negativity for such subsystems in $AdS_3/CFT_2$ scenario is given by

$$\mathcal{E}(A:B) = \mathcal{X}_2 E_W(\rho_{AB}), \tag{29}$$

with $\mathcal{X}_2 = 3/2$, as obtained from eq. (28).

On a related note, in [68] the authors have investigated a stronger bound for an inequality satisfied by the field theoretic dual of the minimal EWCS, namely the reflected entropy [37] in the usual $AdS/CFT$ framework. It was shown in [68] that the difference between the reflected entropy and the holographic mutual information was bounded from below by the fidelity of a Markov recovery process related to the purification of the mixed state under consideration. This difference, termed the Markov gap, possessed an interesting geometric interpretation in terms of the number of non-trivial boundaries of the EWCS which turned out to be a constant at the leading order. Also note that in several earlier holographic proposals [19–21], the entanglement negativity was shown to be proportional to holographic mutual informations between various combinations of the subsystems involved. Hence, we expect that the above proposal in eq. (29) connecting the holographic entanglement negativity with the minimal EWCS receives further correction in terms of the aforementioned Markov gap. This quantity was also explored in [48] in the context of the canonical purification of a given mixed state, where it was termed the crossing partial entanglement entropy (PEE) and remarkably the computations led to the same constant value as in [68]. Therefore, we speculate that a more correct version of the proposal in eq. (29) should include the signature of the Markov gap.

## 3.2 EWCS and entanglement negativity in flat space holography

In our work, however, we are dealing with cases where taking the subsystems at a constant time slice does not give the correct entanglement structure. So we use a non static version of the EWCS given by the area of the bulk codimension-2 *extremal* surface $\Sigma_{AB}^{\text{extr}}$ [44]. To this end, we now explore the connection between the extremal EWCS and the holographic entanglement negativity for the $GCFT_{1+1}$ scenarios dual to the bulk (2+1)-dimensional asymptotically flat gravity theories. In this context, in Appendix A, we consider a generic bipartite mixed state of two disjoint intervals in $CFT_{1+1}$s dual to the bulk $AdS_3$ geometries and perform a parametric İnönü-Wigner contraction of the EWCS computed for this configuration in [35, 44]. Through this parametric contraction we obtain an expression for the EWCS for the non relativistic scenario of bulk asymptotically flat spacetimes. We find that the EWCS for flat gravity scenarios is proportional to the conformal block in $t$-channel where the dominant contribution comes from the composite twist operator $\Phi_{n_e}^2$. In [64], the monodromy analysis for the holographic entanglement negativity for the same configuration discussed above in the context of the bulk asymptotically flat gravity was performed. They found that the associated 4-point function corresponds to the same conformal block we encountered in our computation of EWCS. Comparing the two results one could establish a relation between the EWCS and the holographic entanglement negativity in the context of flat space holography as follows

$$\mathcal{E} = \frac{3}{2} E_W. \tag{30}$$

Although the above proposal connecting a geometric quantity inside the entanglement wedge to a mixed state entanglement measure looks promising, as discussed earlier in the context of $AdS_3/CFT_2$ scenario, the works in [68] calls attention to some modifications in terms of the Markov gap. As described in [56, 64] the entanglement negativity between two subsystems in a $GCFT_{1+1}$ may be expressed in terms of the holographic mutual information between

various combinations of the subsystems involved. Therefore, given a proper definition of the reflected entropy in such $GCFT_{1+1}$s and its putative duality with the EWCS, it is expected that a similar Markov gap will be involved in the comparison of the EWCS and the holographic mutual information.

It should be noted here that in the context of $AdS_3/CFT_2$, the holographic entanglement negativity for a mixed state of two disjoint intervals exhibits a phase transition [30, 32] while going from the $s$-channel where its value is trivially zero to the $t$-channel where the dominant contribution to the conformal block comes from the aforementioned non-trivial twist operator $\Phi_{n_e}^2$. We anticipate a similar phase structure for the flat holographic scenarios as well. We expect that this transition from a zero to a non-zero value of entanglement negativity corresponds in the bulk to the transition from a disconnected entanglement wedge for which the EWCS is zero to a connected entanglement wedge where we get a non-trivial EWCS[3].

We now provide further reasoning in support of our proposal in eq. (30). We use the fact that the HRT prescription used to compute the extremal EWCS where we extremize the area of the corresponding bulk codimension-2 surface is equivalent to the maximin construction [41] where we choose the maximum among multiple minimal area surfaces on the achronal Cauchy slice homologous to the subsystem in question in the dual $CFT_d$ [44]. This equivalence between the HRT construction and the maximin construction for non static $AdS_{d+1}/CFT_d$ scenarios was demonstrated in [41].

Now it should be noted here that the holographic construction used in [60,61] to obtain the holographic entanglement entropy[4] for subsystems in asymptotically flat geometries mimics the HRT construction for non-static bulk $AdS$ geometries. And since a HRT-maximin equivalence exists in the non-static $AdS_{d+1}/CFT_d$ scenarios, we anticipate that a similar equivalent maximin construction should also exist for asymptotically flat geometries. So a maximin version of the covariant holographic entanglement negativity introduced in [27–29] (where the HRT construction has been used) is expected to exist for flat geometries as well. The covariant holographic entanglement negativity in $AdS_3/CFT_2$ scenario has been found to be related to the extremal EWCS through a generic factor of 3/2 [39]. On the basis of above mentioned equivalences for the $AdS_{d+1}/CFT_d$ scenarios, the same relation between the maximin EWCS and the covariant holographic entanglement negativity obtained using the maximin construction should hold in $AdS_3/CFT_2$ framework. Now, this should translate in the flat limit without any modification as the factor of $\mathcal{X}_d$ given in eq. (28) depends only on the bulk dimension and the geometry of the entangling surface which are not affected while taking the flat limit. This serves as a further confirmation of our proposal in eq. (30) for flat space holography.

## 4 EWCS in flat Einstein gravity

Having established a connection between the EWCS and the holographic entanglement negativity for asymptotically flat geometries in (1+1)-dimensions, we now proceed to describe our novel construction for the entanglement wedge for the asymptotically flat background geometry described by the Einstein gravity dual to a $GCFT_{1+1}$. To this end, we consider two intervals $A$ and $B$ with zero overlap in the dual $GCFT_{1+1}$. According to the flat-holographic version of the HRT formula in [61, 64] the holographic entanglement entropy for this configuration is given by the length of the extremal curves $\Gamma_A^{\text{extr}}$, $\Gamma_B^{\text{extr}}$ and $\Gamma_{AB}^{\text{extr}}$ as shown in figure 3 and 4.

We now follow the covariant prescription in [44] to define the entanglement wedge as the

---

[3]This matching between the phase structure of the entanglement negativity and the transition from a disconnected to a connected entanglement wedge was observed for the $AdS/CFT$ scenarios in [11, 13, 49].

[4]The holographic construction used in [64] to compute the entanglement negativity also mimics the HRT construction.

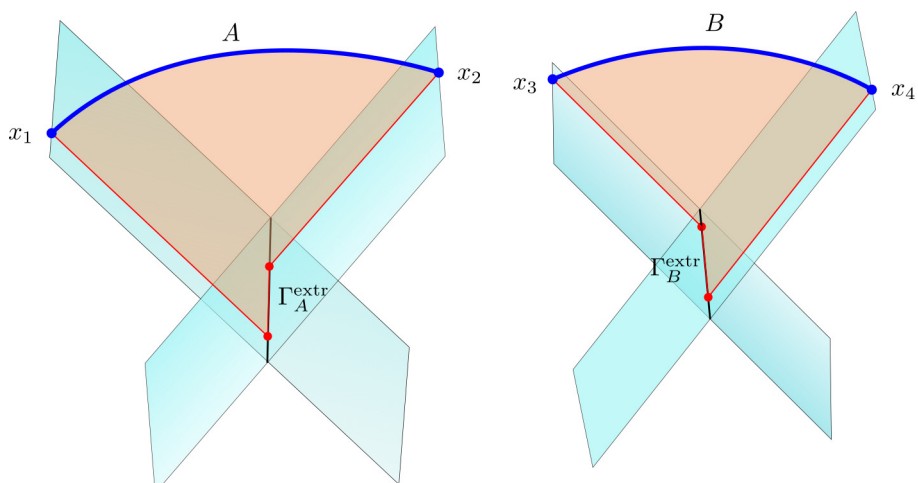

Figure 3: Two disjoint intervals $A$ and $B$ in a $GCFT_{1+1}$ dual to Einstein gravity in asymptotically flat spacetime, such that their entanglement wedge remains disconnected and correspondingly the EWCS is identically zero.

codimension-1 region of the spacetime bounded by the union of the extremal curves which give the dominant contribution to the covariant entanglement entropy. When the size of the intervals $A$ and $B$ are very small such that the covariant entanglement entropy is dominated by the combination of the HRT like curves $\Gamma_A^{\text{extr}}$ and $\Gamma_B^{\text{extr}}$ (figure 3), we get a disconnected entanglement wedge as a codimension-1 surface bounded by the union of curves

$$M_{AB} = A \cup B \cup \Gamma_A^{\text{extr}} \cup \Gamma_B^{\text{extr}}. \tag{31}$$

However when the intervals are big enough such that the covariant entanglement entropy gets a dominant contribution from the HRT like curve $\Gamma_{AB}^{\text{extr}}$ (figure 4), then a connected entanglement wedge is formed and is bounded by the curves

$$M_{AB} = A \cup B \cup \Gamma_{AB}^{\text{extr}}. \tag{32}$$

As described in [44], for the case of a connected entanglement wedge we divide the extremal curve $\Gamma_{AB}^{\text{extr}}$ into two parts as follows

$$\Gamma_{AB}^{\text{extr}} = \Gamma_{AB}^{(A)} \cup \Gamma_{AB}^{(B)}. \tag{33}$$

The extremal entanglement wedge cross section is then defined in terms of the length of the extremal curve $\Sigma_{AB}^{\text{extr}}$ as

$$E_W(\rho_{AB}) = \min_{\Gamma_{AB}^{(A)} \subset \Gamma_{AB}^{\text{extr}}} \text{extr} \left[ \frac{L\left(\Sigma_{AB}\right)}{4 G_N} \right], \tag{34}$$

where the extremization is performed with respect to the division (33), and we have minimized over the possibility of more than one extremal surface.

We will now describe an explicit construction of the entanglement wedge cross section in flat holography for two Galilean boosted disjoint intervals $A = [(x_1, t_1), (x_2, t_2)]$ and $B = [(x_3, t_3), (x_4, t_4)]$ on the boundary. We consider the intervals to have a connected entanglement wedge to get a non-trivial entanglement wedge cross-section. In figure 4, the shaded region bounded by the intervals at the boundary and the extremal curves homologous to the complementary subregions is the corresponding entanglement wedge of the density matrix $\rho_{AB}$. Before proceeding, we specify some properties of the extremal curves homologous to $A$ and $B$. The bulk points $y_i$s lie on the null planes $N_i$s descending from the endpoints of

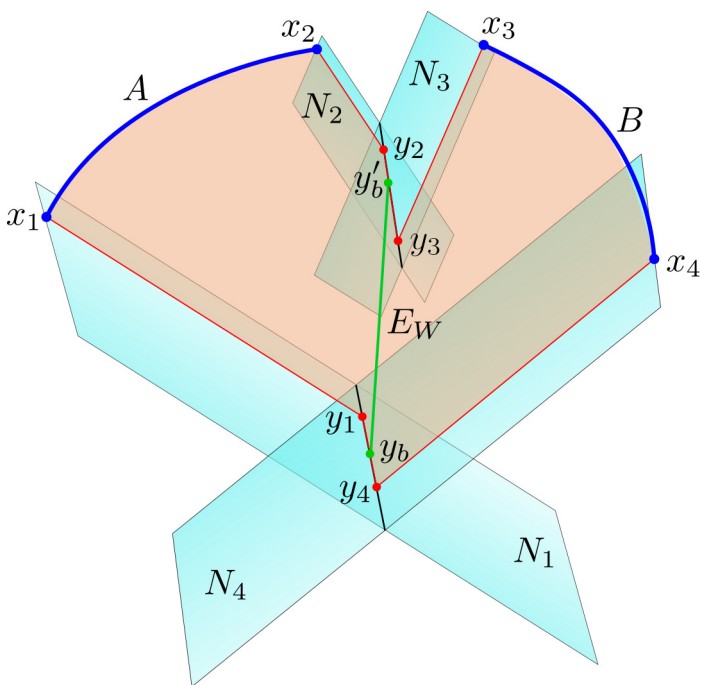

Figure 4: Schematics for the holographic construction of the EWCS for two disjoint intervals in a $GCFT_{1+1}$ dual to Einstein gravity in asymptotically flat spacetime. The shaded region bounded by the red line segments describing the extremal curves $\Gamma_{AB}^{\text{extr}}$ and the boundary intervals $A$ and $B$, is the entanglement wedge dual to $\rho_{AB}$.

the intervals $A$ and $B$ on the boundary. Further, the points $y_1$ and $y_4$ lie on the intersection of the null planes $N_1$ and $N_4$ and the corresponding extremal curve connecting them is denoted by $\gamma_{14}$. Similarly the points $y_2$ and $y_3$ lie on $N_2 \cap N_3$ and the extremal curve connecting them is denoted by $\gamma_{23}$. Since all these curves have been extremized, all the bulk geodesics are straight lines [61]. We now consider two arbitrary bulk points $y_b$ and $y'_b$ on the extremal curves $\gamma_{14}$ and $\gamma_{23}$ respectively. The extremal entanglement wedge cross-section (EWCS) is then obtained by extremizing over the position of these two arbitrary bulk points and is given as

$$E_W = \min_{\substack{y_b \in \gamma_{14} \\ y'_b \in \gamma_{23}}} \text{extr} \left[ \frac{L(y_b, y'_b)}{4G_N} \right]. \tag{35}$$

We now proceed towards the computation of the EWCS for different cases of two disjoint intervals in the $GCFT_{1+1}$ vacuum, for a $GCFT_{1+1}$ describing a finite sized system and for a $GCFT_{1+1}$ at a finite temperature in subsection 4.1. Subsequently, in subsection 4.2 we propose a similar holographic construction for the case two adjacent intervals and compute the EWCS for all the above mentioned $GCFT_{1+1}$ configurations. Further in subsection 4.3 we compute the EWCS for the pure state configuration of a single interval in the $GCFT_{1+1}$ vacuum and for a $GCFT_{1+1}$ describing a finite sized system. For the mixed state configuration of a single interval in a thermal $GCFT_{1+1}$ we obtain the holographic entanglement negativity by utilizing a flat holographic version of the prescription advanced in [38].

## 4.1 EWCS for Two Disjoint Intervals in Proximity

We first consider the mixed state configuration described by two disjoint intervals in proximity in the $GCFT_{1+1}$ dual to (2+1)-dimensional Einstein gravity in asymptotically flat spacetime.

In the following subsections we compute the EWCS for the configurations involving a $GCFT_{1+1}$ in its ground state, a thermal $GCFT_{1+1}$ and a $GCFT_{1+1}$ describing a finite sized system on an infinite cylinder.

### 4.1.1 Two disjoint intervals in vacuum

In this subsection we compute the EWCS for the bipartite mixed state configuration of two disjoint intervals in the $GCFT_{1+1}$ vacuum dual to the bulk flat $(2+1)$-dimensional Minkowski spacetime in Eddington-Finkelstein coordinates which is given as

$$ds^2 = -du^2 + dr^2 + r^2 d\phi^2,\tag{36}$$

where $u = t - r$ is the (retarded) Eddington-Finkelstein time, $r$ is the holographic coordinate, and $\phi$ is the usual angular coordinate with $\phi \sim \phi + 2\pi$.

To this end, we consider the setup of two disjoint intervals $A = [(u_1, \phi_1), (u_2, \phi_2)]$ and $B = [(u_3, \phi_3), (u_4, \phi_4)]$, in the cylindrical coordinates, which are related to the usual planar coordinates at the boundary via eq. (5). The entanglement wedge of the mixed state configuration $A \cup B$ is the shaded bulk codimension$-1$ region in figure 4 and the length of the extremal curve connecting the geodesics $\gamma_{14}$ and $\gamma_{23}$ gives the EWCS.

We now choose the bulk points $y_b \in \gamma_{14}$ with coordinates $(r_b, u_b, \phi_b)$ and $y'_b \in \gamma_{23}$ with coordinates $(r'_b, u'_b, \phi'_b)$. We do so by introducing the parameters

$$\begin{aligned}
s_k &= r_b \sin(\phi_b - \phi_k) && \text{for } k = 1, 4, \\
s_k &= r'_b \sin(\phi'_b - \phi_k) && \text{for } k = 2, 3,
\end{aligned}\tag{37}$$

where $s_1$ and $s_4$ correspond to the lengths on either sides of $y_b$ along $\gamma_{14}$ and $s_2$ and $s_3$ correspond to lengths on either sides of $y'_b$ along $\gamma_{23}$. On utilizing the expression for the length of the extremal curve between two points given in eq. (14), and the constraint that the parameters must add up to the geodesic length, we get

$$\begin{aligned}
s_1 + s_4 &= u_{41} \cot \frac{\phi_{14}}{2}, \\
s_2 + s_3 &= u_{32} \cot \frac{\phi_{23}}{2}.
\end{aligned}\tag{38}$$

Note that the bulk point $y_b$ and $y'_b$ lives on the intersection of the null planes $N_1 \cap N_4$ and $N_2 \cap N_3$, respectively. This further gives us the following constraining equations

$$\begin{aligned}
u_b - u_i + 2r_b \sin^2\left(\frac{\phi_b - \phi_i}{2}\right) &= 0, \quad i = 1, 4, \\
u'_b - u_i + 2r'_b \sin^2\left(\frac{\phi'_b - \phi_i}{2}\right) &= 0, \quad i = 2, 3.
\end{aligned}\tag{39}$$

Using eqs. (37), (38) and (39), we can compute the coordinates of $y_b$ in terms of the known boundary coordinates $(u_i, \phi_i)$ and the parameters $s_4$ as [61]

$$\begin{aligned}
x_b = r_b \cos \phi_b &= \frac{u_{14} \cos \phi_4 + s_4(\sin \phi_1 - \sin \phi_4)}{1 - \cos \phi_{14}}, \\
y_b = r_b \sin \phi_b &= \frac{u_{14} \sin \phi_4 + s_4(\cos \phi_4 - \cos \phi_1)}{1 - \cos \phi_{14}}, \\
t_b = u_b + r_b &= \frac{u_1 - u_4 \cos \phi_{14} + s_4 \sin \phi_{14}}{1 - \cos \phi_{14}}.
\end{aligned}\tag{40}$$

And similar expressions for the coordinates of $y'_b$ can be obtained in terms of $(u_i, \phi_i)$ and the parameter $s_2$. The extremal length between $y_b$ and $y'_b$ may then be obtained by extremizing over the undetermined variables $s_2$ and $s_4$ to be

$$L^{\text{extr}}(y_b, y'_b) = \left| \frac{u_{23} \sin \phi_{41} + u_{24} \sin \phi_{13} + u_{12} \sin \phi_{43} + u_{43} \sin \phi_{12} + u_{31} \sin \phi_{42} + u_{14} \sin \phi_{32}}{8 \sin \frac{\phi_{32}}{2} \sin \frac{\phi_{14}}{2} \sqrt{\sin \frac{\phi_{13}}{2} \sin \frac{\phi_{12}}{2} \sin \frac{\phi_{43}}{2} \sin \frac{\phi_{42}}{2}}} \right|. \quad (41)$$

In our usual planar coordinates at the boundary in eq. (5), the above expression may conveniently be put in terms of the $GCFT_{1+1}$ cross ratios as[5]

$$L^{\text{extr}}(y_b, y'_b) = \left| \frac{X}{\sqrt{T}(1-T)} \right|, \quad (42)$$

where the cross ratios $T$ and $X/T$ are as defined in eq. (11). In the $t$-channel $T \to 1$, the two disjoint intervals are in proximity, and we get the EWCS from eq. (35) as

$$E_W = \frac{1}{4G_N} \frac{X}{1-T} = \frac{1}{4G} \left( \frac{x_{13}}{t_{13}} + \frac{x_{24}}{t_{24}} - \frac{x_{14}}{t_{14}} - \frac{x_{23}}{t_{23}} \right). \quad (43)$$

Finally, substituting this result into our proposal in eq. (30), we obtain the holographic entanglement negativity for the two disjoint intervals in proximity as

$$\mathcal{E} = \frac{c_M}{8} \left( \frac{x_{13}}{t_{13}} + \frac{x_{24}}{t_{24}} - \frac{x_{14}}{t_{14}} - \frac{x_{23}}{t_{23}} \right), \quad (44)$$

where we have used the fact that for bulk Einstein gravity, the central charges in the dual field theory are as given in eq. (13). The expression for the holographic entanglement negativity in eq. (44) matches exactly with the result obtained in [64] using large $c_M$ analysis and through a particular linear combination of the lengths of bulk extremal curves. This serves as a strong consistency check for our holographic construction for the EWCS.

We note in passing that the connection between the holographic entanglement negativity and the extremal EWCS is exact in this case which leads us to speculate that the Markov gap or the crossing PEE is vanishing. We reiterate that a proper understanding of this behaviour requires a careful analysis of such quantities in the flat holographic setting.

### 4.1.2 Two disjoint intervals at a finite temperature

In this subsection we perform the computation of the extremal EWCS for two disjoint intervals in a thermal $GCFT_{1+1}$ defined on a compactified cylinder of circumference equal to the inverse temperature $\beta$. The dual asymptotically flat geometry is given by a non-rotating Flat Space Cosmological (FSC) solution described by the metric given in eq. (16). The temperature in the dual field theory at null infinity is related to the ADM mass of the spacetime as in eq. (17).

As before we consider two disjoint intervals described by $A = [(u_1, \phi_1), (u_2, \phi_2)]$ and $B = [(u_3, \phi_3), (u_4, \phi_4)]$. In this case the analogue of eq. (37) is given by

$$\begin{aligned} s_k &= \frac{r_b}{\sqrt{M}} \sinh\left( \sqrt{M}(\phi_b - \phi_k) \right) \qquad \text{for } k = 1, 4, \\ s_k &= \frac{r'_b}{\sqrt{M}} \sinh\left( \sqrt{M}(\phi'_b - \phi_k) \right) \qquad \text{for } k = 2, 3, \end{aligned} \quad (45)$$

---

[5]Compare this extremal length with eq. (9) to see the proportionality with the conformal block. Also look at [64] for comparison with the conformal block giving the dominant contribution in the entanglement negativity computation.

which are constrained to add up to the lengths of the bulk extremal curves on which the bulk points $y_b$ and $y_b'$ lives. As before the two points $y_b$ and $y_b'$ are chosen to be indigenous to the intersection of the null planes $N_1 \cap N_4$ and $N_2 \cap N_3$, respectively which gives further constraints similar to the ones given in eq. (39). On utilizing these constraints we may now obtain the coordinates of the point $y_b$ in terms of the boundary coordinates $(u_i, \phi_i)$ and the auxiliary parameter $s_4$ as follows

$$
\begin{aligned}
x_b &= \frac{\sqrt{M} u_{14} \sinh\left(\sqrt{M}\phi_4\right) + s_4\left(\cosh\left(\sqrt{M}\phi_1\right) - \cosh\left(\sqrt{M}\phi_4\right)\right)}{\cosh\left(\sqrt{M}\phi_{14}\right) - 1}, \\
t_b &= \frac{\sqrt{M} u_{14} \cosh\left(\sqrt{M}\phi_4\right) + s_4\left(\sinh\left(\sqrt{M}\phi_1\right) - \sinh\left(\sqrt{M}\phi_4\right)\right)}{\cosh\left(\sqrt{M}\phi_{14}\right) - 1}, \\
y_b &= \frac{-\sqrt{M} u_1 + \sqrt{M} u_4 \cosh\left(\sqrt{M}\phi_{14}\right) - s_4 \sinh\left(\sqrt{M}\phi_{14}\right)}{\cosh\left(\sqrt{M}\phi_{14}\right) - 1}.
\end{aligned}
\tag{46}
$$

Similarly the coordinates of $y_b'$ may be expressed in terms of the boundary coordinates and the auxiliary variable $s_2$. The extremal length between $y_b$ and $y_b'$ may now be computed by extremizing over the variables $s_2$ and $s_4$ to obtain

$$
L_{\text{FSC}}^{\text{extr}}(y_b, y_b') = \frac{A}{B},
\tag{47}
$$

where $A$ and $B$ are given by

$$
A = \frac{\sqrt{M}}{8} \left| u_{23} \sinh\left(\sqrt{M}\phi_{41}\right) + u_{24}\sinh\left(\sqrt{M}\phi_{13}\right) + u_{12}\sinh\left(\sqrt{M}\phi_{43}\right) + u_{43}\sinh\left(\sqrt{M}\phi_{12}\right) \right.
$$
$$
\left. + u_{31}\sinh\left(\sqrt{M}\phi_{42}\right) + u_{14}\sinh\left(\sqrt{M}\phi_{32}\right) \right|,
\tag{48a}
$$

$$
B = \left| \sinh\frac{\sqrt{M}\phi_{32}}{2}\sinh\frac{\sqrt{M}\phi_{14}}{2}\sqrt{\sinh\frac{\sqrt{M}\phi_{13}}{2}\sinh\frac{\sqrt{M}\phi_{12}}{2}\sinh\frac{\sqrt{M}\phi_{43}}{2}\sinh\frac{\sqrt{M}\phi_{42}}{2}} \right|.
\tag{48b}
$$

In the $t$-channel ($T \to 1$), the EWCS for two disjoint intervals in proximity in a thermal $GCFT_{1+1}$ may then be obtained using eq. (35) as

$$
E_W = \frac{\pi}{4G_N\beta} \left[ u_{13}\coth\left(\frac{\pi\phi_{13}}{\beta}\right) + u_{24}\coth\left(\frac{\pi\phi_{24}}{\beta}\right) - u_{14}\coth\left(\frac{\pi\phi_{14}}{\beta}\right) - u_{23}\coth\left(\frac{\pi\phi_{23}}{\beta}\right) \right],
\tag{49}
$$

where we have used the relation between the ADM mass of the spacetime and the temperature of the dual field theory $\sqrt{M} = \frac{2\pi}{\beta}$. Utilizing the cross ratios for the finite temperature $GCFT_{1+1}$ given by

$$
\tilde{T} = \frac{\sinh\frac{\sqrt{M}\phi_{12}}{2}\sinh\frac{\sqrt{M}\phi_{34}}{2}}{\sinh\frac{\sqrt{M}\phi_{13}}{2}\sinh\frac{\sqrt{M}\phi_{24}}{2}},
\tag{50a}
$$

$$
\frac{\tilde{X}}{\tilde{T}} = \frac{1}{2}\left( \frac{\sqrt{M}u_{12}}{\tanh\frac{\sqrt{M}\phi_{12}}{2}} + \frac{\sqrt{M}u_{34}}{\tanh\frac{\sqrt{M}\phi_{34}}{2}} - \frac{\sqrt{M}u_{13}}{\tanh\frac{\sqrt{M}\phi_{13}}{2}} - \frac{\sqrt{M}u_{24}}{\tanh\frac{\sqrt{M}\phi_{24}}{2}} \right),
\tag{50b}
$$

eq. (49) may again be expressed in terms of the cross ratios as

$$
E_W = \frac{1}{4G_N}\frac{\tilde{X}}{1 - \tilde{T}}.
\tag{51}
$$

On substituting the EWCS into our proposal in eq. (30), we obtain the expression for the entanglement negativity for disjoint intervals in a thermal $GCFT_{1+1}$ as

$$\mathcal{E} = \frac{c_M \pi}{8\beta} \left[ u_{13} \coth\left(\frac{\pi\phi_{13}}{\beta}\right) + u_{24} \coth\left(\frac{\pi\phi_{24}}{\beta}\right) - u_{14} \coth\left(\frac{\pi\phi_{14}}{\beta}\right) - u_{23} \coth\left(\frac{\pi\phi_{23}}{\beta}\right) \right], \quad (52)$$

where we have used eq. (13). This matches exactly with the results in [64] obtained through an alternate holographic prescription involving a particular linear combination of the lengths of the bulk extremal curves. It should be noted here that again the Markov gap or the crossing PEE vanishes for this case indicating a full Markov recovery.

### 4.1.3 Two disjoint intervals in a finite sized system

Next we focus on a $GCFT_{1+1}$ compactified on a spatial circle of circumference $L_\phi$. The dual geometry is described by the global Minkowski orbifold, which may be obtained from the pure Minkowski spacetime through quotienting with a spatial circle compactified as

$$(u, \phi) \sim (u, \phi + L_\phi). \quad (53)$$

The metric for global Minkowski orbifolds is given in eq. (19). Comparing with eq. (16), it is easy to see that the ADM mass of the dual spacetime is related to the size of the boundary system as given in eq. (20). Therefore, all our previous analysis of subsection 4.1.2 will simply follow with $\sqrt{M} = \frac{2\pi i}{L_\phi}$, and will lead to the expression of EWCS as follows

$$E_W = \frac{\pi}{4G_N L_\phi} \left[ u_{13} \cot\left(\frac{\pi\phi_{13}}{L_\phi}\right) + u_{24} \cot\left(\frac{\pi\phi_{24}}{L_\phi}\right) - u_{14} \cot\left(\frac{\pi\phi_{14}}{L_\phi}\right) - u_{23} \cot\left(\frac{\pi\phi_{23}}{L_\phi}\right) \right]. \quad (54)$$

Using our proposal in eq. (30), we can now get the entanglement negativity for disjoint intervals in a finite sized system as

$$\mathcal{E} = \frac{c_M \pi}{8L_\phi} \left[ u_{13} \cot\left(\frac{\pi\phi_{13}}{L_\phi}\right) + u_{24} \cot\left(\frac{\pi\phi_{24}}{L_\phi}\right) - u_{14} \cot\left(\frac{\pi\phi_{14}}{L_\phi}\right) - u_{23} \cot\left(\frac{\pi\phi_{23}}{L_\phi}\right) \right], \quad (55)$$

where we have made use of eq. (13). This matches exactly with the results obtained in [64]. Again as earlier the connection between the holographic entanglement negativity and the extremal EWCS given in eq. (30) is exact here and we observe that the Markov gap or the crossing PEE vanishes.

## 4.2 EWCS for Two Adjacent Intervals

In this subsection we advance a holographic construction for computing the entanglement wedge cross section for two adjacent subsystems in a $GCFT_{1+1}$ dual to the Einstein gravity in asymptotically flat spacetimes. To this end, we consider two adjacent intervals $A$ and $B$, at the null infinity of a $(2+1)$-dimensional asymptotically flat spacetime. In figure 5, the shaded region is the connected entanglement wedge for this configuration. To compute the EWCS we choose a bulk point $y'_b$ on the extremal curve $\gamma_{13}$ computing the entanglement entropy of the composite system $A \cup B$, and extremize the length between the boundary point $\partial_2 A \equiv \partial_1 B$ and the bulk point $y'_b$. Note that in our construction the bulk point $y'_b$ must lie on the intersection of the null planes $N_1$ and $N_3$ descending from the endpoints of $A \cup B$. However, as described in [61], the distance from a bulk point to the boundary on the future infinity is still infinite and we need a prescription to regulate the length. To this end, we consider an arbitrary point $y_b$ on a null line $\gamma_2$ descending from the boundary point $\partial_2 A$ as our regulator. The regulated distance between the boundary point $\partial_2 A$ and $y'_b$ may now be written as

$$L(\partial_2 A, y'_b) = L(\partial_2 A, y_b) + L(y_b, y'_b) = L(y_b, y'_b). \quad (56)$$

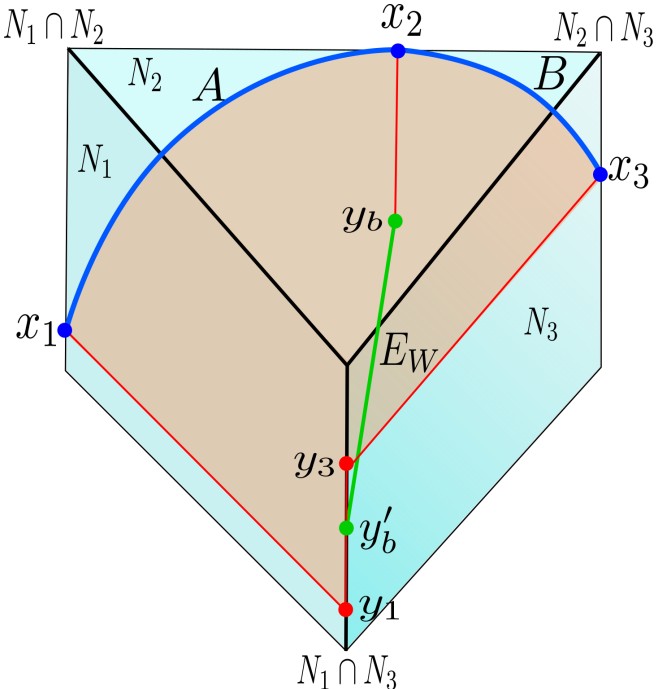

Figure 5: Schematics of the holographic construction for computing the extremal EWCS for two adjacent intervals in a $GCFT_{1+1}$ dual to Einstein gravity in asymptotically flat spacetime.

Note that the position of $y_b$ along the null line is arbitrary, so we must extremize the length $L(y_b, y'_b)$ with respect to the positions of both the bulk points $y_b$ and $y'_b$. Therefore we may express the correct EWCS for the configuration of two adjacent intervals as

$$E_W^{\text{adj}} = \min_{\substack{y'_b \in \gamma_{13} \\ y_b \in \gamma_2}} \text{extr} \left[ \frac{L(y_b, y'_b)}{4G_N} \right].$$ (57)

Note that although the above construction is perfectly valid when one starts with the setup of two adjacent intervals in the dual field theory at the asymptotic boundary, it should be consistent with a suitable adjacent limit of the proposal involving two disjoint intervals in eq. (35). To this end, in the limit of $x_2 \to x_3$, the extremal curve $\gamma_{23}$ in the construction for two disjoint intervals described in section 4 shrinks to a single point $\gamma_{23} \equiv y_b^{\text{extr}}$. However we would like to emphasize that upon extremization over the free parameter labelled $y_b$ in fig. 5, it is precisely identical to the point $y_b^{\text{extr}}$. Hence the extremization procedure described above consistently leads to an identical construction to the one arrived at from a limiting analysis of the corresponding disjoint configuration. However, if one begins with an independent setup of two adjacent intervals, the location of the point $y_b^{\text{extr}}$ in the bulk cannot be evaluated directly and can only be fixed after the extremization procedure described above[6].

We now proceed to compute the extremal EWCS for the three cases of a $GCFT_{1+1}$ in its ground state, a thermal $GCFT_{1+1}$ and a $GCFT_{1+1}$ describing a finite sized system on an infinite cylinder as considered before in section 4.1.

### 4.2.1 Two adjacent intervals in vacuum

In this subsection we proceed to compute the extremal EWCS for two adjacent intervals $A = [(u_1, \phi_1), (u_2, \phi_2)]$ and $B = [(u_2, \phi_2), (u_3, \phi_3)]$ in the vacuum state of a $GCFT_{1+1}$ dual

---

[6]We would like to thank the referee for pointing out this subtlety.

to Einstein gravity in Minkowski spacetime. As described above, we drop a null curve from $(u_2, \phi_2)$ till an arbitrary bulk point $y_b$ with the bulk coordinate $r_b$ as the unspecified parameter. The coordinates of $y_b$ may be written in terms of the coordinates on the boundary as

$$x_b = r_b \cos \phi_2, \quad y_b = r_b \sin \phi_2, \quad t_b = u_2 + r_b. \tag{58}$$

To express the position of the bulk point $y_b'$, living on $N_1 \cap N_3$, in terms of the boundary coordinates we again employ the method described in subsection 4.1.1 and introduce two parameters describing the geodesic length on either side of $y_b'$ as

$$s_1 = r_b' \sin(\phi_b' - \phi_1), \quad s_3 = r_b' \sin(\phi_b' - \phi_3), \tag{59}$$

where

$$s_1 + s_3 = u_{31} \cot \frac{\phi_{13}}{2}. \tag{60}$$

Since the bulk point $y_b'$ lies on the intersection of the null planes $N_1$ and $N_3$, its coordinates must obey the constraint equations given as

$$u_b' - u_i + 2r_b' \sin^2\left(\frac{\phi_b' - \phi_i}{2}\right) = 0, \quad i = 1, 3. \tag{61}$$

Utilizing these constraints in eq. (60) and (61), we may obtain the following expressions for the position of $y_b'$,

$$
\begin{aligned}
x_b' &= \frac{u_{13} \cos \phi_3 + s_3(\sin \phi_1 - \sin \phi_3)}{1 - \cos \phi_{13}}, \\
y_b' &= \frac{u_{13} \sin \phi_3 + s_3(\cos \phi_3 - \cos \phi_1)}{1 - \cos \phi_{13}}, \\
t_b' &= \frac{u_1 - u_3 \cos \phi_{13} + s_3 \sin \phi_{13}}{1 - \cos \phi_{13}}.
\end{aligned}
\tag{62}
$$

The extremal length required for the computation of the EWCS may now be obtained by extremizing the length $L(y_b, y_b')$, with respect to the undetermined variables $r_b$ and $s_3$ to be

$$L^{\text{extr}}(y_b, y_b') = \frac{1}{2}\left( \frac{u_{12}}{\tan \frac{\phi_{12}}{2}} + \frac{u_{23}}{\tan \frac{\phi_{23}}{2}} - \frac{u_{13}}{\tan \frac{\phi_{13}}{2}} \right). \tag{63}$$

Therefore, the extremal EWCS may be expressed in the planar coordinates in eq. (5) as

$$E_W^{\text{adj}} = \frac{1}{4G_N}\left( \frac{x_{12}}{t_{12}} + \frac{x_{23}}{t_{23}} - \frac{x_{13}}{t_{13}} \right). \tag{64}$$

We provide a consistency check of the above computation by taking the adjacent limit of the disjoint result in eq. (43). In terms of the boundary coordinates this corresponds to taking the limit $x_{23} \to \epsilon$, $t_{23} \to \tilde{\epsilon}$ with $\epsilon, \tilde{\epsilon} \to 0$[7]. This modifies the cross-ratio given in eq. (11). Putting this modified cross-ratio back in eq. (43), we obtain an expression which exactly matches the explicit holographic computation for adjacent intervals in eq. (64). Finally, we utilize our proposal in eq. (30) to obtain the holographic entanglement negativity for the mixed state configuration of two adjacent intervals in vacuum to be

$$\mathcal{E} = \frac{c_M}{8}\left( \frac{x_{12}}{t_{12}} + \frac{x_{23}}{t_{23}} - \frac{x_{13}}{t_{13}} \right). \tag{65}$$

---

[7] We would like to emphasize that in order to have a finite entanglement negativity, one should really subtract divergent terms of $\mathcal{O}(\frac{\epsilon}{\tilde{\epsilon}})$ from each of the terms of form $\frac{x}{t}$ which appear in expressions of entanglement entropies in [60]. In this manuscript we chose to omit such divergent terms from the general expressions for brevity. For this reason, we consistently neglect the $\mathcal{O}(\frac{\epsilon}{\tilde{\epsilon}})$ term coming from $\frac{x_{23}}{t_{23}}$ in eq. (43) while implementing the adjacent limit.

This matches exactly with the dual field theory result for $c_L = 0$ in [56] and with that in [64] obtained through another equivalent holographic construction utilizing a certain algebraic sum of the extremal curves. This serves as a strong consistency check of our construction. Interestingly, we find a vanishing Markov gap or the crossing PEE for this configuration as well and complete understanding of the implication of this feature requires a proper analysis of the Markov recovery process in asymptotically flat spacetimes.

### 4.2.2 Two adjacent intervals at a finite temperature

In this subsection we focus on a mixed state configuration of two adjacent intervals $A = [(u_1, \phi_1), (u_2, \phi_2)]$ and $B = [(u_2, \phi_2), (u_3, \phi_3)]$ in a thermal $GCFT_{1+1}$ dual to the non-rotating FSC solution described by the metric given in eq. (16). As mentioned earlier, the inverse temperature $\beta$ in the dual field theory at the null infinity is related to the ADM mass $M$ of the spacetime as given in eq. (17).

As per our construction given in subsection 4.2 (see figure 5), we need the extremal length $L^{\text{extr}}(y_b, y_b')$ between the bulk point $y_b'$ and the point $y_b$ acting as a regulator to obtain the EWCS. We can retrace our computation of the previous subsection of defining two generic parameters describing the geodesic length on either side of $y_b'$ as follows

$$s_k = \frac{r_b'}{\sqrt{M}} \sinh\left(\sqrt{M}(\phi_b' - \phi_k)\right) \quad \text{for } k = 1, 3 . \tag{66}$$

Naturally they add up to the length of the bulk extremal curve $\gamma_{13}$ on which $y_b'$ lives. Also the bulk point $y_b'$ is constrained to lie on the intersection of the null planes $N_1 \cap N_3$ which is given by

$$M(u_b' - u_i) + 2r_b' \sinh^2\left(\frac{\sqrt{M}(\phi_b' - \phi_i)}{2}\right) = 0 , \quad i = 1, 3 . \tag{67}$$

On utilizing these constraints, similar to the previous subsection, we are left with two undetermined variables in the expression of the length $L(y_b, y_b')$. We may extremize over these variables to obtain the extremal length as follows

$$L^{\text{extr}}(y_b, y_b') = \frac{\pi u_{12}}{\beta} \coth\left(\frac{\pi \phi_{12}}{\beta}\right) + \frac{\pi u_{23}}{\beta} \coth\left(\frac{\pi \phi_{23}}{\beta}\right) - \frac{\pi u_{13}}{\beta} \coth\left(\frac{\pi \phi_{13}}{\beta}\right) , \tag{68}$$

where eq. (17) has been used. Using the above expression, we may compute the expression of the EWCS for the mixed state configuration in question as

$$E_W = \frac{1}{4G_N} \left[ \frac{\pi u_{12}}{\beta} \coth\left(\frac{\pi \phi_{12}}{\beta}\right) + \frac{\pi u_{23}}{\beta} \coth\left(\frac{\pi \phi_{23}}{\beta}\right) - \frac{\pi u_{13}}{\beta} \coth\left(\frac{\pi \phi_{13}}{\beta}\right) \right] . \tag{69}$$

The consistency of the above result may be checked by taking the adjacent limit of the disjoint result in eq. (49). This corresponds to taking the limit $x_{23} \to \epsilon$, $t_{23} \to \tilde{\epsilon}$ with $\epsilon, \tilde{\epsilon} \to 0$ which modifies the cross-ratios given in eq. (50). Using these modified cross-ratios we obtain an expression which exactly matches the explicit bulk computation of the EWCS for adjacent intervals in eq. (69). Further, our proposal in eq. (30) may be utilized to obtain the holographic entanglement negativity for the two adjacent intervals in the mixed state configuration in question as

$$\mathcal{E} = \frac{c_M}{8} \left[ \frac{\pi u_{12}}{\beta} \coth\left(\frac{\pi \phi_{12}}{\beta}\right) + \frac{\pi u_{23}}{\beta} \coth\left(\frac{\pi \phi_{23}}{\beta}\right) - \frac{\pi u_{13}}{\beta} \coth\left(\frac{\pi \phi_{13}}{\beta}\right) \right] , \tag{70}$$

where eq. (13) is used. Again this matches exactly for $c_L = 0$ with the result obtained in [56] through a dual field theory analysis and with that in [64] obtained through another equivalent holographic construction utilizing a certain algebraic sum of the extremal curves. It should be noted here that the relation between the holographic entanglement negativity and the EWCS given in eq. (30) is exact here as the Markov gap or the crossing PEE vanishes.

### 4.2.3 Two adjacent intervals in a finite sized system

In this subsection, we compute the EWCS for two adjacent intervals $A = [(u_1, \phi_1), (u_2, \phi_2)]$ and $B = [(u_2, \phi_2), (u_3, \phi_3)]$ in a $GCFT_{1+1}$ compactified on a spatial circle of circumference $L_\phi$. As described earlier, the dual geometry is the global Minkowski orbifold with the metric given in eq. (19) and the ADM mass $M$ of the dual spacetime is related to the size of the boundary system $L_\phi$ as given in eq. (20). Again in a manner similar to the disjoint interval case in subsection 4.1.3, our analysis for finite sized system follows the finite temperature scenario of subsection 4.2.2 with $\sqrt{M} = \frac{2\pi i}{L_\phi}$. This gives us the expression of the EWCS for the adjacent subsystems in question as follows

$$E_W = \frac{1}{4G_N} \left[ \frac{\pi u_{12}}{L_\phi} \cot\left(\frac{\pi \phi_{12}}{L_\phi}\right) + \frac{\pi u_{23}}{L_\phi} \cot\left(\frac{\pi \phi_{23}}{L_\phi}\right) - \frac{\pi u_{13}}{L_\phi} \cot\left(\frac{\pi \phi_{13}}{L_\phi}\right) \right]. \tag{71}$$

Like earlier the above result may also be verified by taking the adjacent limit of the disjoint result in eq. (54). Now using our proposal in eq. (30), we may obtain the expression for the holographic entanglement negativity for two adjacent subsystems in a $GCFT_{1+1}$ compactified on a spatial circle as

$$\mathcal{E} = \frac{c_M}{8} \left[ \frac{\pi u_{12}}{L_\phi} \cot\left(\frac{\pi \phi_{12}}{L_\phi}\right) + \frac{\pi u_{23}}{L_\phi} \cot\left(\frac{\pi \phi_{23}}{L_\phi}\right) - \frac{\pi u_{13}}{L_\phi} \cot\left(\frac{\pi \phi_{13}}{L_\phi}\right) \right], \tag{72}$$

which matches exactly with the result in [56, 64] for $c_L = 0$. Again, the Markov gap or the crossing PEE for this configuration is vanishing and we speculate full Markov recovery.

## 4.3 EWCS for a Single Interval

Finally we consider the pure state configuration of a single interval in a $GCFT_{1+1}$ vacuum and in a $GCFT_{1+1}$ describing a finite sized system, and a mixed state configuration of a single interval in a thermal $GCFT_{1+1}$. For the pure state scenario the EWCS is given trivially by the length of the extremal curve homologous to the subsystem on the boundary [44]. But for the mixed state in a thermal $GCFT_{1+1}$, the construction is subtle. We utilize here a flat holographic version of the construction used in [38] to obtain the EWCS for the $AdS_3/CFT_2$ framework.

### 4.3.1 Single interval in vacuum

Consider a pure bipartite state given by an interval $A = [(u_1, \phi_1), (u_2, \phi_2)]$ and its compliment $B = A^c$ in a $GCFT_{1+1}$ in the vacuum state with the endpoints of $A$ being labelled as $\partial_i A$, see figure 1. The dual geometry is the Einstein gravity in the Minkowski spacetime. In subsection 2.2, we saw that flat holographic version of the HRT curve for this configuration is comprised of the null segments $\gamma_i$ starting at $\partial_i A$ connected via a third extremal bulk geodesic.

Now, for the EWCS for the pure state configuration of a single interval at zero temperature, we require the length of the extremal geodesic homologous to the interval on the boundary given in eq. (14). So, we may get the expression for entanglement wedge cross-section as

$$E_W = \frac{1}{4G_N} L_A^{\text{extr}} = \frac{1}{4G_N} \left| \frac{u_{21}}{\tan\frac{\phi_{12}}{2}} \right|, \tag{73}$$

which obviously is same as the holographic entanglement entropy of the interval $A$, as expected for a pure state. Substituting in our proposal (30), we may obtain the corresponding holographic entanglement negativity in planar coordinates, given in eq. (5), to be

$$\mathcal{E} = \frac{c_M}{4} \frac{x_{12}}{t_{12}}. \tag{74}$$

This is precisely the result obtained in [56, 64] for $c_L = 0$ which provides another consistency check for our proposal.

### 4.3.2 Single interval in a finite sized system

Consider a single interval $A = [(u_1, \phi_1), (u_2, \phi_2)]$ in a $GCFT_{1+1}$ compactified on a spatial circle of circumference $L_\phi$ dual to the global Minkowski orbifold whose metric is given in eq. (19). As mentioned in earlier subsections, the ADM mass of the dual spacetime is related to the size of the boundary system as given in eq. (20).

For the computation of EWCS we need the length of the extremal bulk curve homologous to the interval $A$ which is given in eq. (21). Utilizing this expression, we may obtain the extremal EWCS for the single interval in a $GCFT_{1+1}$ compactified on a spatial circle as

$$E_W = \frac{1}{4G_N} L_A^{\text{extr}} = \frac{1}{4G_N} \frac{\pi u_{12}}{L_\phi} \cot\left(\frac{\pi \phi_{12}}{L_\phi}\right). \tag{75}$$

Putting the above expression for the EWCS in our proposal (30) gives the expression for the holographic entanglement negativity as follows

$$\mathcal{E} = \frac{c_M}{4} \frac{\pi u_{12}}{L_\phi} \cot\left(\frac{\pi \phi_{12}}{L_\phi}\right), \tag{76}$$

which matches exactly with the results obtained in [56, 64] for $c_L = 0$.

### 4.3.3 Single interval at a finite temperature

Finally we consider the case of a bipartite configuration of interval $A$ and its compliment $B = A^c$ in the $GCFT_{1+1}$ at a finite temperature whose bulk dual is the non-rotating FSC solution with the metric given in eq. (16). The ADM mass of the spacetime $M$ is related to the inverse temperature $\beta$ of the dual field theory at the null infinity as given in eq. (17).

However the construction in this case is subtle as was shown in [38] in the context of $AdS/CFT$. We propose a construction similar to the one mentioned above for the computation of EWCS for a single interval in a thermal $GCFT_{1+1}$ in the context of flat space holography. As described in [56, 64] the mixed state configuration of a single interval $A$ in a finite temperature $GCFT_{1+1}$ is correctly described by sandwiching the single interval in question between two auxiliary large intervals $B_1$ and $B_2$ on both sides. To this end we consider a tripartite system described by $A \cup B_1 \cup B_2$ as shown in figure 6. The length of the single interval $A = [(u_1, \phi_1), (u_2, \phi_2)]$ is denoted by $\ell$ and we choose the lengths of the auxiliary subsystems $B_1$ and $B_2$ to be $L$. Similar to the situation described in [38] in the $AdS_3/CFT_2$ framework, in the bipartite limit $L \to \infty$ the original configuration of a single interval $A$ with the rest of the system given by $B_1 \cup B_2 = A^c$ is recovered. Therefore, we have the following inequalities for tripartite pure state configurations at our disposal [44, 45]:

$$\begin{aligned} E_W(A : B_1 B_2) &\leq E_W(A : B_1) + E_W(A : B_2), \\ E_W(A : B_1 B_2) &\geq \frac{1}{2} I(A : B_1) + \frac{1}{2} I(A : B_2). \end{aligned} \tag{77}$$

Now we specialize to the case of $A$ and $B_1$ (or $B_2$) being adjacent to each other. From eq. (69) and the corresponding entanglement entropy in eq. (18), it is easy to see that the following equality holds:

$$E_W(A : B) = \frac{1}{2} I(A : B). \tag{78}$$

Therefore, the inequalities in (77) reduce to equalities, and in particular we obtain

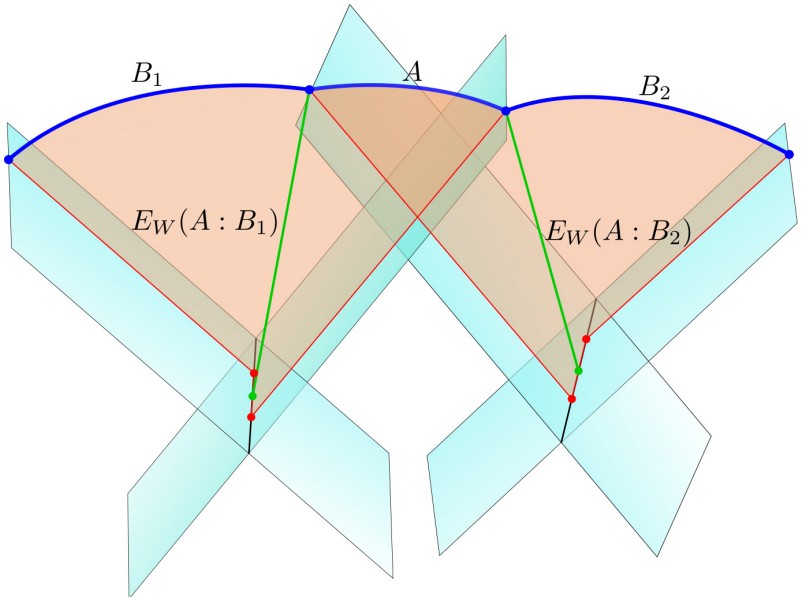

Figure 6: EWCS construction for a single interval in a thermal $GCFT_{1+1}$ dual to the FSC geometry. Note that the EWCS shown by the green curves only provide a schematic while the actual construction for adjacent intervals requires a regularization prescription as discussed in section 4.2.

$$E_W(A : B_1 B_2) = E_W(A : B_1) + E_W(A : B_2), \tag{79}$$

for the present configuration in the bipartite limit.

Therefore, utilizing eqs. (69) and (79) we obtain the extremal EWCS for the bipartite mixed state of a single interval on an infinite cylinder of circumference $\beta$ as

$$E_W(A) = \lim_{L \to \infty} E_W(A : B_1 B_2) = \frac{1}{4 G_N} \left[ \frac{\pi u_{12}}{\beta} \coth\left( \frac{\pi \phi_{12}}{\beta} \right) - \frac{\pi u_{12}}{\beta} \right]. \tag{80}$$

Utilizing our proposal in eq. (30) we may obtain the holographic entanglement negativity for the configuration in question as

$$\mathcal{E} = \frac{c_M}{4} \left[ \frac{\pi u_{12}}{\beta} \coth\left( \frac{\pi \phi_{12}}{\beta} \right) - \frac{\pi u_{12}}{\beta} \right], \tag{81}$$

which exactly matches with the corresponding results in [56, 64]. Note here that our proposal connecting the holographic entanglement negativity and the EWCS in eq. (30) is exact in this case as the Markov gap or the crossing PEE vanishes. This serves as a strong substantiation for our holographic construction which is equivalent to the one given in [64]. Note that eq. (81) could be recast in the instructive form

$$\mathcal{E} = \frac{3}{2} \left( S_A - S_A^{\text{th}} \right), \tag{82}$$

where $S_A$ is the entanglement entropy of the interval $A$ on the boundary which may be obtained utilizing eqs. (15) and (18) and $S_A^{\text{th}} = \frac{c_M}{4} \frac{\pi u_{12}}{\beta}$ denotes the thermal contribution. This subtraction of the thermal contribution may be interpreted as the entanglement negativity providing the upper bound of distillable entanglement.

# 5 EWCS in flat-space TMG

In the previous sections we established the holographic constructions for the extremal EWCS in asymptotically flat geometries described by Einsten gravity. From the point of view of the dual field theory, this corresponds to a generic bipartite density matrix $\rho_{AB}$ in a $GCFT_{1+1}$ with only one non-vanishing central charge $c_M$. In this section, we will lift this restriction and consider the effects of a non-zero $c_L$. The modified bulk picture is described by the flat space Topologically Massive Gravity (TMG). The action for flat space TMG includes a gravitational Chern-Simons (CS) term coupled to the usual Einstein-Hilbert action:

$$S_{\text{flat-TMG}} = \frac{1}{16\pi G} \int d^3x \ \sqrt{-g}\Big[R + \frac{1}{2\mu}\varepsilon^{\alpha\beta\gamma}\Big(\Gamma^{\rho}_{\alpha\sigma}\partial_\beta\Gamma^{\sigma}_{\gamma\rho} + \frac{2}{3}\Gamma^{\rho}_{\alpha\sigma}\Gamma^{\sigma}_{\beta\eta}\Gamma^{\eta}_{\gamma\rho}\Big)\Big], \qquad (83)$$

where $\mu$ is the coupling of the CS term with the Einstein-Hilbert action. When the coupling is weak ($\mu \to \infty$), the TMG action reduces to the Einstein gravity. Note that the flat-space TMG may be obtained as a flat limit $\ell \to \infty$ of the familiar $AdS$-TMG action, where $\ell$ is the $AdS_3$ radius [64]. The asymptotic symmetry analysis at null infinity of flat-space TMG yields the Galilean conformal algebra with non-trivial central extensions:

$$c_L = \frac{3}{\mu G_N}, \quad c_M = \frac{3}{G_N} \, . \qquad (84)$$

As described in [61,64], a non-vanishing $c_L$ in the dual $GCFT_{1+1}$ corresponds to primary operators with nontrivial spin. The holographic correspondence in asymptotically flat spacetimes dictates that such operators in the dual field theory correspond to massive spinning particles propagating in the bulk. The on-shell worldline action for such an anyonic particle of mass $\chi$ and spin $\Delta$ was found in [61,75] to be [8]

$$S_{\text{on-shell}} = \int_C ds \left( \chi \sqrt{\eta_{\mu\nu}\dot{X}^\mu \dot{X}^\nu} + \Delta \, (\tilde{n}.\nabla n) \right), \qquad (85)$$

where $\tilde{n}$ and $n$ are unit spacelike and timelike vectors normal to the trajectory of the particle $X^\mu$ and $C$ denotes the worldline of the particle. The Chern-Simons term in eq. (83) effectively erects a normal frame $(X, n, \tilde{n})$ to each point in the bulk, thereby modifying the shape of particle worldline in the form of a ribbon. The Chern-Simons contribution in eq. (85) may be interpreted as the boost required to drag the normal frame $(X, n, \tilde{n})$ from the point $x_i$ to $x_f$ [61]:

$$\Delta\eta(n_i, n_f) = \int_C ds \, (\tilde{n}.\nabla n) = \cosh^{-1}(-n_i.n_f), \qquad (86)$$

where $n_i$ and $n_f$ are the normal vectors at $x_i$ and $x_f$, respectively. A natural realization for the regulated bulk normal timelike vectors on the null geodesics $\gamma_i$ descending from the boundary point $(u_i, \phi_i)$ was described in [61] by introducing an auxiliary null vector $m_i$ as

$$n_i = \frac{1}{\tilde{\epsilon}}\dot{\gamma}_i + \tilde{\epsilon} m_i \, , \qquad (87)$$

where $\gamma_i.m_i = -1$ and $\tilde{\epsilon}$ is a UV cut-off. It was also demonstrated that the on-shell action computed using the above prescription was independent of the regulator $\tilde{\epsilon}$. In the following we will utilize this prescription to propose a holographic construction to obtain the Chern-Simons contibution to the extremal EWCS corresponding to a generic bipartite state in the dual $GCFT_{1+1}$.

---

[8]Note that $\chi$ , $\Delta$ are the scaling dimensions of the corresponding primary operator in the dual $GCFT_{1+1}$ as described in section 2.1.

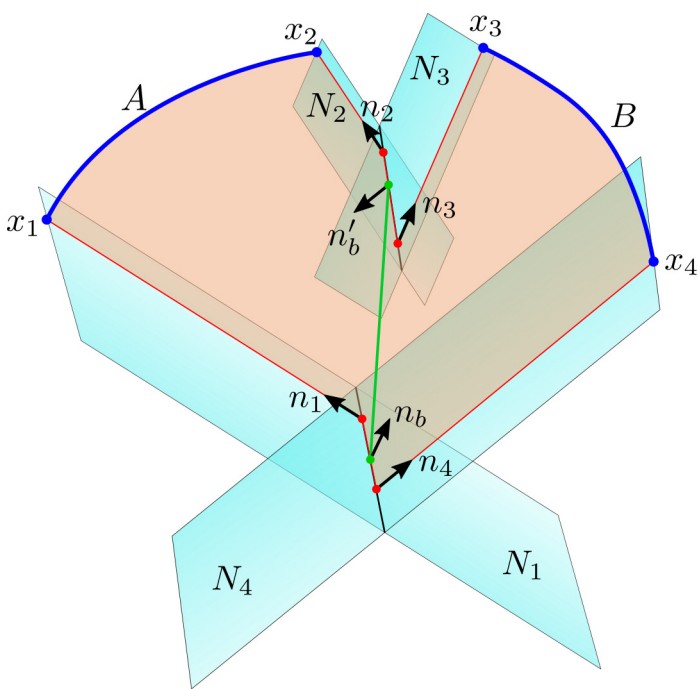

Figure 7: Schematics of the holographic construction of extremal EWCS for two disjoint intervals in a $GCFT_{1+1}$ dual to flat-space TMG.

Consider two generic intervals $A$ and $B$ with no overlap in the dual $GCFT_{1+1}$. The construction of the entanglement wedge for this configuration is identical to that discussed in section 4 with an extra ingredient that there are timelike vectors erected at each bulk point, as shown in figure 7. The Chern-Simons contribution to the extremal EWCS is obtained by extremizing the boost required to drag the normal frame through the bulk entanglement wedge. In particular we extremize over the relative orientation of the bulk timelike vectors $n_b$ and $n'_b$ located at the bulk points $y_b$ and $y'_b$ which were obtained in the construction of the EWCS for the flat Einstein gravity in section 4.1. Therefore the Chern-Simons contribution to the extremal EWCS is given by

$$E_W^{\text{CS}} = \min_{n_b, n'_b} \text{extr} \left[ \frac{\Delta \eta(n_b, n'_b)}{4\mu G_N} \right]. \tag{88}$$

Note that in our definition of the extremal EWCS the coupling constant $\mu$ for the Chern-Simons term appears in the denominator which fixes the EWCS to be dimensionless.

In the following subsections, we will compute the above CS contribution to the extremal EWCS for various bipartite states $\rho_{AB}$ in the dual $GCFT_{1+1}$ and subsequently obtain the holographic entanglement negativity utilizing our proposal in eq. (30) generalized to include the Chern-Simons contribution. Note that, as discussed earlier in subsection 3.2, the connection between the holographic entanglement negativity and the EWCS involves the appearance of a Markov gap which turned out to be vanishing in the case of bulk Einstein gravity in asymptotically flat spacetimes. In the present scenario, however, we will find that the Chern-Simons contribution to the EWCS differs from half the holographic mutual information signifying a non-trivial Markov gap or crossing PEE. Therefore, the holographic proposal for the entanglement negativity in eq. (30) requires suitable modifications to incorporate the Markov gap. Fortunately, as in the $AdS_3/CFT_2$ scenario described in [68], the Markov gap will turn out to be a constant and therefore the functional structure of our holographic proposal in eq. (30) remains unchanged.

To begin with, we consider the bipartite mixed state of two disjoint intervals in the

$GCFT_{1+1}$ vacuum, in a thermal $GCFT_{1+1}$ and subsequently for a $GCFT_{1+1}$ describing a finite sized system. Next we compute the extremal EWCS for two adjacent intervals from a specific bulk construction. These results may also be obtained from the disjoint interval results by taking an appropriate limit. The computations for the pure state configurations described by a single interval in the ground state of a $GCFT_{1+1}$, and for a finite sized system described by a $GCFT_{1+1}$ on an infinite cylinder will follow the same spirit of subsection 4.3. Finally, we will describe single interval in a thermal $GCFT_{1+1}$ defined on an infinite cylinder of circumference $\beta$ compactified in the timelike direction. In this case, we will modify our construction in subsection (4.3.3) to include the effects of a non-zero $c_L$ and the EWCS will be obtained using the bipartite limit of the auxilary intervals.

## 5.1 EWCS for two disjoint intervals

In this subsection, we will calculate the Chern-Simons contribution to the holographic entanglement negativity through the extremal EWCS for two disjoint intervals in the vacuum state, at a finite temperature, and for finite sized systems in dual $GCFT_{1+1}$s. In this context, we discuss the bulk construction for the EWCS in flat-space TMG background for these configurations which involves bulk vectors at each bulk point. In particular the Chern-Simons contribution for the EWCS contains, the scalar product of two bulk timelike vectors $n_b$ and $n_b'$. These vectors are placed on the endpoints of the extremal EWCS as constructed in the Einstein gravity case. Our results for the holographic entanglement negativity thus obtained exactly match with the corresponding expressions in [64].

### 5.1.1 Two disjoint intervals in vacuum

We start with TMG in Minkowski spacetime. A schematics of the bulk geometry corresponding to two disjoint intervals $A = [(u_1, \phi_1), (u_2, \phi_2)]$ and $B = [(u_3, \phi_3), (u_4, \phi_4)]$ in the dual $GCFT_{1+1}$ is shown in figure 7. We have bulk normal vectors $n_i$ at each of the bulk points $y_i$ descending from the endpoints $(u_i, \phi_i)$ of the intervals on the boundary, which were chosen in [61] to be pointed along the directions of the corresponding null geodesics $\gamma_i$:

$$\dot{\gamma}_i = \partial_r \Big|_{\gamma_i} = \partial_t + \cos\phi_i\,\partial_x + \sin\phi_i\,\partial_y \quad (i = 1,..,4). \tag{89}$$

As described earlier, we also need two arbitrary timelike vectors $n_b$ and $n_b'$ at the points $y_b$ and $y_b'$ landing on the intersections of the null planes as shown in figure 7. In order to compute the Chern-Simons contribution to the extremal EWCS, we need to extremize the total boost $\Delta\eta$ given in eq. (86). A convenient parametrization of the bulk timelike normal vectors was given in [61] which we reproduce here:

$$\begin{aligned}
n_b &= \frac{x^2 + u^2 + 1}{2u}\partial_t + \frac{x^2 + u^2 - 1}{2u}\partial_x + \frac{x}{u}\partial_y\,, \\
n_b' &= \frac{x'^2 + u'^2 + 1}{2u'}\partial_t + \frac{x'^2 + u'^2 - 1}{2u'}\partial_x + \frac{x'}{u'}\partial_y\,,
\end{aligned} \tag{90}$$

where $x$, $x'$ and $u$, $u'$ are free parameters. We now use the freedom provided by Galilean conformal symmetry to move the boundary end points at the symmetric locations

$$\phi_{1,4} = \pi \mp \frac{\Phi}{2}\,, \quad \text{and} \quad \phi_{2,3} = \pm\frac{\Phi}{2}\,. \tag{91}$$

After fixing the endpoints of the intervals on the boundary as above, the boosts along the extremal curves on either sides of $y_b$ (or $y_b'$) are given by $\mathcal{L}\sin\frac{\Phi}{2}$ [61], where $\mathcal{L}$ is arbitrary.

These constraints allow us to find the total boost in eq. (86) as

$$\Delta\eta = 2\log\left[\cot\left(\frac{\Phi}{4}\right)\frac{\mathcal{L}\pm\sqrt{\mathcal{L}^2-4}}{2}\right].$$

(92)

Extremization with respect to $\mathcal{L}$ leads to an additive constant which is divergent and may be regularized by adding a counterterm in the total action. The final extremized boost after regularization is given by

$$\Delta\eta^{\text{extr}} = 2\log\left[\cot\frac{\Phi}{4}\right].$$

(93)

In terms of the cross-ratio (11), eq. (93) may be conveniently rewritten as

$$\Delta\eta = 2\log\left[\frac{1+\sqrt{T}}{\sqrt{1-T}}\right].$$

(94)

For the two disjoint intervals in proximity ($T \to 1$), the above equation reduces to

$$\Delta\eta \approx 2\log 2 - \log(1-T).$$

(95)

Therefore, the Chern-Simons contribution to the extremal EWCS for the two disjoint intervals in proximity may be explicitly written in terms of the planar coordinates in eq. (5) as

$$E_W^{\text{CS}} = \frac{1}{4\mu G_N}\log\left(\frac{t_{13}t_{24}}{t_{14}t_{23}}\right) + \frac{1}{4\mu G_N}\log 4.$$

(96)

We note that the constant term on the right hand side of the above expression may be interpreted as the Markov gap or the crossing PEE for the mixed state of two disjoint intervals in vacuum. In this case, we observe that the constant contribution may be obtained as $\frac{\log 2}{4\mu G_N}$ times the number of boundaries of the EWCS. As a consequence, we expect the geometric interpretation of the Markov gap as described in [68] to hold in the framework of flat space holography as well.

Finally, substituting eq. (96) in our proposal (30), we obtain the Chern-Simons contribution to the holographic entanglement negativity for two disjoint intervals in proximity in the ground state of a $GCFT_{1+1}$ to be

$$\mathcal{E}^{\text{CS}} = \frac{c_L}{8}\log\left(\frac{t_{13}t_{24}}{t_{14}t_{23}}\right),$$

(97)

where we have omitted the constant term of the right hand side which carries the signature of the Markov gap or the crossing PEE [48,68]. Interestingly, we did not have a Markov gap in the usual Einstein gravity case as discussed earlier in section 4. It is somewhat counter-intuitive that the topological part of the action captures the Markov gap while the truly dynamical part does not. However a complete understanding of this behaviour requires a proper analysis of the Markov recovery process or the balanced partial entanglement in the context of flat space holography.

The complete expression for the holographic entanglement negativity together with the Einstein gravity result in eq. (44) becomes

$$\mathcal{E} = \frac{c_L}{8}\log\left(\frac{t_{13}t_{24}}{t_{14}t_{23}}\right) + \frac{c_M}{8}\left(\frac{x_{13}}{t_{13}} + \frac{x_{24}}{t_{24}} - \frac{x_{14}}{t_{14}} - \frac{x_{23}}{t_{23}}\right),$$

(98)

which matches exactly with the holographic entanglement negativity obtained in [64] through a particular linear combination of the bulk extremal lengths homologous to specific intervals.

### 5.1.2 Two disjoint intervals at a finite temperature

We now compute the extremal EWCS for the two disjoint intervals $A = [(u_1, \phi_1), (u_2, \phi_2)]$ and $B = [(u_3, \phi_3), (u_4, \phi_4)]$ at a finite temperature. In this context, the dual field theory is defined on an infinite cylinder compactified in the timelike direction and the corresponding bulk theory is described by TMG in FSC geometry. Note that the nature of the timelike vectors at each bulk point changes due to the FSC geometry. The authors in [64] obtained the bulk vector $n_i$ at each bulk point $y_i$ as given in eq. (87), where

$$\dot{\gamma}_i = \frac{1}{\sqrt{M}} \cosh\left(\sqrt{M}\phi_i\right) \partial_t + \frac{1}{\sqrt{M}} \sinh\left(\sqrt{M}\phi_i\right) \partial_x - \frac{1}{\sqrt{M}} \partial_y \quad (i = 1, ..., 4), \tag{99}$$

and $m_i \equiv 0$. In eq. (99), $M$ is the ADM mass of the spacetime which is related to inverse temperature $\beta$ as given in eq. (17). To compute the total boost (86), we use eq. (90) for the bulk timelike vectors [61]. Next we utilize the Galilean conformal symmetry to place the intervals on the boundary at the symmetric positions

$$\phi_{1,4} = \frac{i\pi}{\sqrt{M}} \mp \frac{\Phi}{2}, \quad \text{and} \quad \phi_{2,3} = \pm\frac{\Phi}{2}. \tag{100}$$

With this choice of the angular location, the lengths of the extremal curves on either side of the bulk points $y_b$ and $y_b'$ can be fixed as

$$\begin{aligned}
-2\dot{\gamma}_i \cdot n_b &= \mathcal{L} \sinh\left(\frac{\sqrt{M}\Phi}{2}\right) \quad \text{for} \quad i = 1, 4, \\
-2\dot{\gamma}_i \cdot n_b' &= \mathcal{L} \sinh\left(\frac{\sqrt{M}\Phi}{2}\right) \quad \text{for} \quad i = 2, 3,
\end{aligned} \tag{101}$$

where $\mathcal{L}$ is an arbitary parameter. As described in the previous subsection, we obtain the total extremized boost, utilizing the above constraint equations, as

$$\Delta\eta = 2 \log\left[\coth\left(\frac{\sqrt{M}\Phi}{4}\right)\right]. \tag{102}$$

We can express the above equation in terms of the EWCS using eq. (88) as

$$E_W^{\text{CS}} = \frac{1}{2\mu G_N} \log\left[\coth\left(\frac{\sqrt{M}\Phi}{4}\right)\right]. \tag{103}$$

For two disjoint interval in proximity, the EWCS can be written in terms of the cross-ratio (50) for the finite temperature $GCFT_{1+1}$ as in eq. (95) with $T$ replaced by $\tilde{T}$. Therefore, utilizing eq. (50), the CS contribution to the extremal EWCS for the configuration in question may be expressed as

$$E_W^{\text{CS}} = \frac{1}{4\mu G_N} \log\left[\frac{\sinh\frac{\sqrt{M}\phi_{13}}{2} \sinh\frac{\sqrt{M}\phi_{24}}{2}}{\sinh\frac{\sqrt{M}\phi_{14}}{2} \sinh\frac{\sqrt{M}\phi_{23}}{2}}\right] + \frac{1}{4\mu G_N} \log 4. \tag{104}$$

Now we can obtain the Chern-Simons contribution to the holographic entanglement negativity for two disjoint intervals at a finite temperature by using our proposal eq. (30) as

$$\mathcal{E}^{\text{CS}} = \frac{c_L}{8} \log\left[\frac{\sinh\left(\frac{\pi\phi_{13}}{\beta}\right) \sinh\left(\frac{\pi\phi_{24}}{\beta}\right)}{\sinh\left(\frac{\pi\phi_{14}}{\beta}\right) \sinh\left(\frac{\pi\phi_{23}}{\beta}\right)}\right], \tag{105}$$

where we have used eq. (17) and have omitted the constant term signifying the Markov gap or the crossing PEE. Utilizing eq. (52), the total expression for the holographic entanglement negativity becomes

$$\mathcal{E} = \frac{c_L}{8} \log \left[ \frac{\sinh\left(\frac{\pi\phi_{13}}{\beta}\right)\sinh\left(\frac{\pi\phi_{24}}{\beta}\right)}{\sinh\left(\frac{\pi\phi_{14}}{\beta}\right)\sinh\left(\frac{\pi\phi_{23}}{\beta}\right)} \right] + \frac{c_M \pi}{8\beta}\left[ u_{13}\coth\left(\frac{\pi\phi_{13}}{\beta}\right) + u_{24}\coth\left(\frac{\pi\phi_{24}}{\beta}\right) \right.$$
$$\left. - u_{14}\coth\left(\frac{\pi\phi_{14}}{\beta}\right) - u_{23}\coth\left(\frac{\pi\phi_{23}}{\beta}\right) \right]. \tag{106}$$

The above equation exactly matches with the holographic entanglement negativity for two disjoint intervals at a finite temperature [64].

### 5.1.3 Two disjoint intervals in a finite-sized system

Finally we compute the Chern-Simons contribution to the EWCS for the two disjoint intervals in a finite sized system. The dual field theory is described on infinite cylinder compactified in the spatial direction with circumference $L_\phi$ and the corresponding bulk theory is described by TMG in global Minkowski orbifold. In this context, the computation of the EWCS is similar to the earlier case but here the ADM mass $M$ is related to the size of the boundary system $L_\phi$ as given in eq. (20). Thus the EWCS for this case becomes

$$E_W^{\mathrm{CS}} = \frac{1}{4\mu G_N} \log \left[ \frac{\sin\left(\frac{\pi\phi_{13}}{L_\phi}\right)\sin\left(\frac{\pi\phi_{24}}{L_\phi}\right)}{\sin\left(\frac{\pi\phi_{14}}{L_\phi}\right)\sin\left(\frac{\pi\phi_{23}}{L_\phi}\right)} \right] + \frac{1}{4\mu G_N} \log 4. \tag{107}$$

Using our proposal (30), the Chern-Simons contribution to the holographic entanglement negativity for two disjoint intervals in a finite sized system may be obtained as

$$\mathcal{E}^{\mathrm{CS}} = \frac{c_L}{8} \log \left[ \frac{\sin\left(\frac{\pi\phi_{13}}{L_\phi}\right)\sin\left(\frac{\pi\phi_{24}}{L_\phi}\right)}{\sin\left(\frac{\pi\phi_{14}}{L_\phi}\right)\sin\left(\frac{\pi\phi_{23}}{L_\phi}\right)} \right], \tag{108}$$

where the constant term in eq. (107) carrying the signature of the Markov gap or the crossing PEE has been omitted. Therefore, the total holographic entanglement negativity using eq. (55) becomes

$$\mathcal{E} = \frac{c_L}{8} \log \left[ \frac{\sin\left(\frac{\pi\phi_{13}}{L_\phi}\right)\sin\left(\frac{\pi\phi_{24}}{L_\phi}\right)}{\sin\left(\frac{\pi\phi_{14}}{L_\phi}\right)\sin\left(\frac{\pi\phi_{23}}{L_\phi}\right)} \right] + \frac{c_M \pi}{8 L_\phi}\left[ u_{13}\cot\left(\frac{\pi\phi_{13}}{L_\phi}\right) + u_{24}\cot\left(\frac{\pi\phi_{24}}{L_\phi}\right) \right.$$
$$\left. - u_{14}\cot\left(\frac{\pi\phi_{14}}{L_\phi}\right) - u_{23}\cot\left(\frac{\pi\phi_{23}}{L_\phi}\right) \right]. \tag{109}$$

The above equation exactly matches with the holographic entanglement negativity results obtained in [64].

## 5.2 EWCS for two adjacent intervals

The bulk construction of EWCS for the adjacent intervals in $GCFT_{1+1}$ is similar to the Einstein gravity case with timelike vectors erected at the each bulk points as shown in figure 8. The computation of the EWCS involves two regulated bulk vectors $n_b$, $n_b'$ placed on the bulk points $y_b$, $y_b'$ which lie respectively on the null line $\gamma_2$ and the intersection of the null planes $N_1 \cap N_3$.

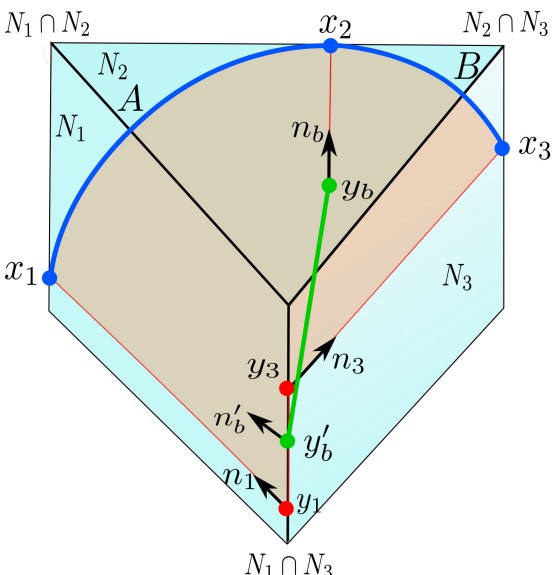

Figure 8: Schematics of the holographic construction of extremal EWCS for two adjacent intervals in a $GCFT_{1+1}$ dual to flat-space TMG

Therefore the EWCS for this configuration is given by

$$E_W^{\text{CS}} = \min_{n_b, n_b'} \text{extr} \left[ \frac{\Delta \eta(n_b, n_b')}{4 \mu G_N} \right].$$
(110)

In the following, we will compute the above Chern-Simons contribution to EWCS for two adjacent intervals in various configurations involving a $GCFT_{1+1}$ vacuum, a $GCFT_{1+1}$ at finite temperature, and a $GCFT_{1+1}$ describing a finite sized system.

### 5.2.1 Two adjacent intervals in vacuum

We start with the computation of Chern-Simons contribution to EWCS for two adjacent intervals $A = [(u_1, \phi_1), (u_2, \phi_2)]$ and $B = [(u_2, \phi_2), (u_3, \phi_3)]$ in the ground state of a $GCFT_{1+1}$. The bulk dual is described by TMG in pure Minkowski space. In this context, the bulk vectors can be obtained using eq. (89) and eq. (90) as

$$n_b = \frac{\dot{\gamma}_2}{\tilde{\epsilon}},$$
$$n_b' = \frac{x^2 + u^2 + 1}{2u} \partial_t + \frac{x^2 + u^2 - 1}{2u} \partial_x + \frac{x}{u} \partial_y,$$
(111)

where $x$, $u$ are arbitrary parameters and $\tilde{\epsilon}$ is the UV cut-off as described before. The choice of these bulk vectors comes from the construction of EWCS described above. Now we can use the freedom provided by the Galilean conformal symmetry to move the angular points to symmetric positions

$$\phi_1 = \pi - \frac{\Phi}{2}, \quad \phi_2 = 0, \quad \phi_3 = \pi + \frac{\Phi}{2}.$$
(112)

With this choice of the location of the boundary points, we can fix the length of the extremal curve to either side of bulk point $y_b'$ as

$$-2\dot{\gamma}_1 \cdot n_b = \mathcal{L} \sin\left(\frac{\Phi}{2}\right),$$
$$-2\dot{\gamma}_3 \cdot n_b' = \mathcal{L} \sin\left(\frac{\Phi}{2}\right),$$
(113)

where $\dot{\gamma}_i$s are null vectors located at the bulk points $y_1$, $y_3$ and $\mathcal{L}$ is an arbitrary parameter. Using above constraint equations, we can now calculate the total boost in (110) utilizing the definition in eq. (86) as

$$\Delta\eta = \log\left[\frac{1}{2\tilde{\epsilon}}\cot\left(\frac{\Phi}{4}\right)\right] - \log\left[\frac{\mathcal{L}\pm\sqrt{\mathcal{L}^2-4}}{8}\right].\tag{114}$$

Next we extremize the boost over the parameter $\mathcal{L}$ which again gives a divergent result. As described earlier, this can be regularized by adding a counter term in the total action. Therefore the Chern-Simons contribution to EWCS using eq. (110) becomes

$$E_W^{\text{CS}} = \frac{1}{4\mu G_N}\log\left[\frac{1}{2\tilde{\epsilon}}\cot\left(\frac{\Phi}{4}\right)\right].\tag{115}$$

The above expression can be restructured in terms of the $GCFT_{1+1}$ cross-ratios using eq. (8). This may further be rewritten in the planar coordinates in eq. (5) as

$$E_W^{\text{CS}} = \frac{1}{4\mu G_N}\log\left[\frac{t_{12}t_{23}}{\tilde{\epsilon}(t_{12}+t_{23})}\right] + \frac{1}{4\mu G_N}\log 2.\tag{116}$$

Once again, we note that the constant term on the right hand side in eq. (116) carries the signature of the Markov gap for the mixed state configuration under consideration. Remarkably, this Markov gap may again be obtained in terms of the number of non-trivial boundaries of the EWCS and we anticipate the above geometric interpretation to hold in generic flat holographic setups.

Finally, we can compute the Chern-Simons contribution to the holographic entanglement negativity for adjacent interval in vacuum by utilizing eqs. (30) and (84) as

$$\mathcal{E}^{\text{CS}} = \frac{c_L}{8}\log\left[\frac{t_{12}t_{23}}{\tilde{\epsilon}(t_{12}+t_{23})}\right],\tag{117}$$

where the constant term signifying the Markov gap or the crossing PEE has been omitted. As a consistency check, we can also obtain eq. (117) from the disjoint intervals result in eq. (97) by taking the appropriate adjacent limit. In particular, this corresponds to $x_{23}\to\epsilon$, $t_{23}\to\tilde{\epsilon}$ with $\epsilon,\tilde{\epsilon}\to 0$ in terms of the planar coordinates. The Chern-Simons contribution to the holographic entanglement negativity obtained through this limit exactly matches with eq. (117).

The total expression for the holographic entanglement negativity may be obtained by including the Eintein gravity contribution in eq. (65) as

$$\mathcal{E} = \frac{c_L}{8}\log\left[\frac{t_{12}t_{23}}{\tilde{\epsilon}(t_{12}+t_{23})}\right] + \frac{c_M}{8}\left(\frac{x_{12}}{t_{12}}+\frac{x_{23}}{t_{23}}-\frac{x_{13}}{t_{13}}\right),\tag{118}$$

which exactly matches with the corresponding results obtained in [56,64].

### 5.2.2 Two adjacent intervals at a finite temperature

Next we compute the Chern-Simons contribution to the EWCS for two adjacent intervals $A = [(u_1,\phi_1),(u_2,\phi_2)]$ and $B = [(u_2,\phi_2),(u_3,\phi_3)]$ at a finite temperature. The boundary theory is defined on an infinite cylinder compactified in the timelike direction and the dual gravitational theory is given by TMG in FSC geometry. In this context, the computation of EWCS involves two bulk vectors $n_b$ and $n_b'$ which are given in eq. (111) with the null vector $\dot{\gamma}_2$ in eq. (99). Now we can utilize the Galilean conformal symmetry to move the angular co-ordinate to symmetric positions

$$\phi_1 = \frac{i\pi}{\sqrt{M}}-\frac{\Phi}{2}, \quad \phi_2 = 0, \quad \phi_3 = \frac{i\pi}{\sqrt{M}}+\frac{\Phi}{2}.\tag{119}$$

In the above equation, $M$ is the ADM mass of the spacetime and is related to inverse temperature $\beta$ of the dual field theory at the null infinity as given in eq. (17). Here we can fix the portions of the length of the extremal curve $\gamma_{13}$ on either sides of the EWCS as

$$
\begin{aligned}
-2\dot{\gamma}_1 \cdot n_b &= \mathcal{L} \sinh\left(\frac{\sqrt{M}\Phi}{2}\right), \\
-2\dot{\gamma}_3 \cdot n'_b &= \mathcal{L} \sinh\left(\frac{\sqrt{M}\Phi}{2}\right).
\end{aligned}
\tag{120}
$$

We can now obtain the extremal EWCS using eqs. (110) and (120) which involves the calculation of the boost in eq. (86) and subsequently an extremization over the arbitrary parameter $\mathcal{L}$. Thus the EWCS for two adjacent intervals at a finite temperature is given by

$$
E_W^{\text{CS}} = \frac{1}{4\mu G_N} \log\left[\frac{1}{2\tilde{\epsilon}} \coth\left(\frac{\sqrt{M}\Phi}{4}\right)\right].
\tag{121}
$$

The above equation can be rewritten, using eq. (50) for the cross-ratios, as

$$
E_W^{\text{CS}} = \frac{1}{4\mu G_N} \log\left[\frac{\beta}{\pi\tilde{\epsilon}} \frac{\sinh\left(\frac{\pi\phi_{12}}{\beta}\right)\sinh\left(\frac{\pi\phi_{23}}{\beta}\right)}{\sinh\left(\frac{\pi(\phi_{12}+\phi_{23})}{\beta}\right)}\right] + \frac{1}{4\mu G_N} \log 2,
\tag{122}
$$

where we have utilized eq. (17). We may now obtain the Chern-Simons contribution to the holographic entanglement negativity for two adjacent intervals at a finite temperature using eqs. (30) and (84) as

$$
\mathcal{E}^{\text{CS}} = \frac{c_L}{8} \log\left[\frac{\beta}{\pi\tilde{\epsilon}} \frac{\sinh\left(\frac{\pi\phi_{12}}{\beta}\right)\sinh\left(\frac{\pi\phi_{23}}{\beta}\right)}{\sinh\left(\frac{\pi(\phi_{12}+\phi_{23})}{\beta}\right)}\right].
\tag{123}
$$

Note that the constant term on the right hand side in eq. (122) carrying the signature of the Markov gap or the crossing PEE has been omitted. We provide a substantiation for eq. (123) by taking the adjacent limit $x_{23} \to \epsilon$, $t_{23} \to \tilde{\epsilon}$ with $\epsilon$, $\tilde{\epsilon} \to 0$ of the disjoint result in eq. (105). The Chern-Simons contribution to the holographic entanglement negativity thus obtained, exactly matches with eq. (123).

Finally, utilizing the Einstein gravity result in eq. (70), the total expression for the holographic entanglement negativity becomes

$$
\mathcal{E} = \frac{c_L}{8} \log\left[\frac{\beta}{\pi\tilde{\epsilon}} \frac{\sinh\left(\frac{\pi\phi_{12}}{\beta}\right)\sinh\left(\frac{\pi\phi_{23}}{\beta}\right)}{\sinh\left(\frac{\pi(\phi_{12}+\phi_{23})}{\beta}\right)}\right] + \frac{c_M}{8}\left[\frac{\pi u_{12}}{\beta} \coth\left(\frac{\pi\phi_{12}}{\beta}\right) + \frac{\pi u_{23}}{\beta} \coth\left(\frac{\pi\phi_{23}}{\beta}\right)\right.
$$
$$
\left. - \frac{\pi u_{13}}{\beta} \coth\left(\frac{\pi\phi_{13}}{\beta}\right)\right].
\tag{124}
$$

The above expression exactly matches with the results obtained in [56, 64] which serves as strong consistency check of our holographic construction.

### 5.2.3 Two adjacent intervals in a finite sized system

Finally we compute the Chern-Simons contribution to the EWCS for two adjacent intervals in a finite sized system described by a $GCFT_{1+1}$ compactified in the spatial direction with circumference $L_\phi$. The dual gravity theory is described by TMG in the global Minkowski orbifold. To

this end, we follow a similar analysis for the computation of the extremal EWCS as shown in subsection 5.2.2. In particular, we obtain the similar expression for the EWCS in the terms of the cross-ratio eq. (50). Therefore the EWCS is given by

$$E_W^{\text{CS}} = \frac{1}{4\mu G_N} \log\left[\frac{\beta}{\pi\tilde{\epsilon}} \frac{\sin\left(\frac{\pi\phi_{12}}{L_\phi}\right)\sin\left(\frac{\pi\phi_{23}}{L_\phi}\right)}{\sin\left(\frac{\pi(\phi_{12}+\phi_{23})}{L_\phi}\right)}\right] + \frac{1}{4\mu G_N}\log 2, \tag{125}$$

where we have utilized the relation between ADM mass and size of the boundary system as given in eq. (20). Finally, we can obtain Chern-Simons contribution to the holographic entanglement negativity using eqs. (30) and (84) as

$$\mathcal{E}^{\text{CS}} = \frac{c_L}{8} \log\left[\frac{\beta}{\pi\tilde{\epsilon}} \frac{\sin\left(\frac{\pi\phi_{12}}{L_\phi}\right)\sin\left(\frac{\pi\phi_{23}}{L_\phi}\right)}{\sin\left(\frac{\pi(\phi_{12}+\phi_{23})}{L_\phi}\right)}\right], \tag{126}$$

where the constant term on the right hand side indicative of the Markov gap or the crossing PEE has been omitted. As discussed in earlier case, we can also compute the above expression for the Chern-Simons contribution to the holographic entanglement negativity by taking the adjacent limit of the disjoint result eq. (108). The total expression for the holographic entanglement negativity by including Einstein gravity result in eq. (72) becomes

$$\mathcal{E} = \frac{c_L}{8} \log\left[\frac{\beta}{\pi\tilde{\epsilon}} \frac{\sin\left(\frac{\pi\phi_{12}}{L_\phi}\right)\sin\left(\frac{\pi\phi_{23}}{L_\phi}\right)}{\sin\left(\frac{\pi(\phi_{12}+\phi_{23})}{L_\phi}\right)}\right] + \frac{c_M}{8}\left[\frac{\pi u_{12}}{L_\phi}\cot\left(\frac{\pi\phi_{12}}{L_\phi}\right) + \frac{\pi u_{23}}{L_\phi}\cot\left(\frac{\pi\phi_{23}}{L_\phi}\right)\right.$$
$$\left. - \frac{\pi u_{13}}{L_\phi}\cot\left(\frac{\pi\phi_{13}}{L_\phi}\right)\right], \tag{127}$$

which exactly matches with the results obtained in [56,64].

## 5.3 EWCS for a single interval

In this subsection, we first consider the pure state configuration of the single interval described by a $GCFT_{1+1}$ in vacuum and a $GCFT_{1+1}$ describing a finite sized system. The bulk gravitational theory is given by TMG in pure Minkowski spacetime and TMG in global Minkowski orbifold, respectively. In particular for the pure state case, the EWCS is equivalent to the holographic computation of the entanglement entropy. Therefore, as described in [64], the construction of the extremal EWCS involves two bulk regulated timelike vectors $n_1$ and $n_2$ located at the bulk points $y_1$ and $y_2$. Note that the above mentioned bulk timelike vectors lie on the intersection of the null plane $N_1$ and $N_2$ dropping from the endpoints of the interval on the boundary [61]. Therefore, the Chern-Simons contribution to the EWCS is given by

$$E_W^{\text{CS}} = \frac{\Delta\eta(n_1, n_2)}{4\mu G_N}, \tag{128}$$

where $\mu$ is the coupling constant of the Chern-Simons term in the action eq. (83).

In the following, we will compute the Chern-Simons contribution to the extremal EWCS for the pure states described by a single interval in a $GCFT_{1+1}$ at zero temperature and in a $GCFT_{1+1}$ describing a finite sized system using eq. (128). The mixed state configuration of of a single interval in a thermal $GCFT_{1+1}$ requires a more careful analysis. To calculate the EWCS for such a configuration in an infinite system, we will follow the same prescription as in the case with pure Einstein gravity with the additional ingredient that there are regulated timelike vectors at each bulk point.

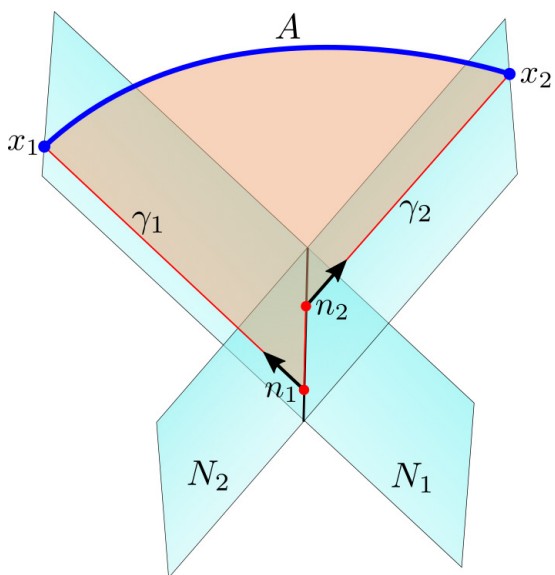

Figure 9: EWCS for the single interval in the flat-space TMG

### 5.3.1 Single interval in vacuum

In the subsection 4.3.1, it was shown that the EWCS for a single interval at zero temperature is solely described by the extremal length in the bulk dual Einstein gravity. In this case, the EWCS for a single interval $A = [(u_1, \phi_1), (u_2, \phi_2)]$ in the ground state of a $GCFT_{1+1}$ dual to TMG in pure Minkowski spacetime can be obtained using boost parameter eq. (86) which involves the scalar product of two timelike vectors $n_1$, $n_2$ located at the bulk points as shown in figure 9. These vectors have the following parametrization

$$
\begin{aligned}
n_1 &= \frac{\dot{\gamma}_1}{\tilde{\epsilon}} - \frac{\tilde{\epsilon}}{2} \frac{\dot{\gamma}_2}{\dot{\gamma}_1 \cdot \dot{\gamma}_2}, \\
n_2 &= \frac{\dot{\gamma}_2}{\tilde{\epsilon}} - \frac{\tilde{\epsilon}}{2} \frac{\dot{\gamma}_1}{\dot{\gamma}_1 \cdot \dot{\gamma}_2},
\end{aligned}
\tag{129}
$$

where $\dot{\gamma}_i$s are the null vectors defined earlier in eq. (89) and $\tilde{\epsilon}$ is the UV cutoff. Using eq. (86), we can compute the boost as

$$
\Delta \eta = 2 \log \left( \frac{2}{\tilde{\epsilon}} \sin \frac{\phi_{12}}{2} \right),
\tag{130}
$$

Thus we can obtain the expression for the extremal EWCS using eqs. (128) and (130) as

$$
E_W^{\mathrm{CS}} = \frac{1}{2 \mu G_N} \log \left( \frac{t_{12}}{\tilde{\epsilon}} \right),
\tag{131}
$$

where we have made use of the planar coordinates in eq. (5). Finally the Chern-Simons contribution to the holographic entanglement negativity for a single interval in a $GCFT_{1+1}$ vacuum may be obtain using our proposal in eq. (30) as

$$
\mathcal{E}^{\mathrm{CS}} = \frac{c_L}{4} \log \left( \frac{t_{12}}{\tilde{\epsilon}} \right).
\tag{132}
$$

In the above equation, we have used the expression for the central charge given in eq. (84). Using eq. (74), the complete expression for the holographic entanglement negativity for the pure state configuration in question becomes

$$
\mathcal{E} = \frac{c_L}{4} \log \left( \frac{t_{12}}{\tilde{\epsilon}} \right) + \frac{c_M}{4} \left( \frac{x_{12}}{t_{12}} \right),
\tag{133}
$$

which exactly matches with the results obtained in [56, 64].

### 5.3.2 Single interval in a finite sized system

Now we compute the Chern-Simons contribution to the extremal EWCS for a single interval $A = [(u_1, \phi_1), (u_2, \phi_2)]$ in a finite sized system. The field theory at the asymptotic boundary is given by a $GCFT_{1+1}$ on an infinite cylinder compactified in the spatial direction with circumference $L_\phi$ and the bulk gravitational dual is described by TMG in the global Minkowski orbifold. The calculation of the boost is similar to the earlier case and can be obtained using eq. (86) as

$$\Delta \eta = 2 \log \left[ \frac{L_\phi}{\pi \tilde{\epsilon}} \sin \left( \frac{\pi \phi_{12}}{L_\phi} \right) \right], \tag{134}$$

where we have used the relation between ADM mass and the size of the boundary system in eq. (20). Utilizing eq. (128), the Chern-Simons contribution to the EWCS becomes

$$E_W^{\text{CS}} = \frac{1}{2 \mu G_N} \log \left[ \frac{L_\phi}{\pi \tilde{\epsilon}} \sin \left( \frac{\pi \phi_{12}}{L_\phi} \right) \right]. \tag{135}$$

Finally, the Chern-Simons contribution to the holographic entanglement negativity for a single interval in a finite sized system may be obtain using our proposal (30) as

$$\mathcal{E}^{\text{CS}} = \frac{c_L}{4} \log \left[ \frac{L_\phi}{\pi \tilde{\epsilon}} \sin \left( \frac{\pi \phi_{12}}{L_\phi} \right) \right], \tag{136}$$

where we have utilized eq. (84) for the central charge $c_L$. Therefore using the Einstein gravity result in eq. (76) the complete expression for the holographic entanglement negativity may be obtained as

$$\mathcal{E} = \frac{c_L}{4} \log \left[ \frac{L_\phi}{\pi \tilde{\epsilon}} \sin \left( \frac{\pi \phi_{12}}{L_\phi} \right) \right] + \frac{c_M}{4} \frac{\pi u_{12}}{L_\phi} \cot \left( \frac{\pi \phi_{12}}{L_\phi} \right), \tag{137}$$

which exactly matches with the results obtained in [56, 64].

### 5.3.3 Single interval at a finite temperature

As discussed earlier in the Einstein gravity scenario, the construction of the EWCS for the mixed state configuration of a single interval at a finite temperature in the $GCFT_{1+1}$ requires a careful analysis. Therefore following the insights forwarded in [38], we utilize the same bulk construction proposed earlier for the Einstein gravity case with additional bulk vectors located at each bulk points. Note that for the tripartite pure state configuration described in subsection 4.3.3, in the present case involving TMG in FSC geometry the inequality in eq. (77) also holds. However, in the present scenario of TMG in asymptotically flat spacetimes, we find from eq. (122) the following equality for two adjacent intervals $A$ and $B$:

$$E_W^{\text{CS}}(A : B) = \frac{1}{2} I^{\text{CS}}(A : B) + \frac{1}{4 \mu G_N} \log 2, \tag{138}$$

where we have utilized results for the Chern-Simons contribution to the entanglement entropies for various subsystems from [61, 64]. Note that, there is a non-perturbative additive constant on the right hand side carrying the signature of the Markov gap (or the crossing PEE) which was absent in eq. (78). Consequently, the reasoning leading to (an analogue of) eq. (79) no longer holds and we may only have an upper bound on the Chern-Simons contribution to the EWCS. Therefore, for a tripartite pure state $A \cup B_1 \cup B_2$, we denote the upper bound by $\tilde{E}_W^{\text{CS}}$ which is given as

$$\tilde{E}_W^{\text{CS}}(A : B_1 B_2) = \frac{1}{2} I^{\text{CS}}(A : B_1) + \frac{1}{2} I^{\text{CS}}(A : B_2) + \frac{1}{4 \mu G_N} \log 4. \tag{139}$$

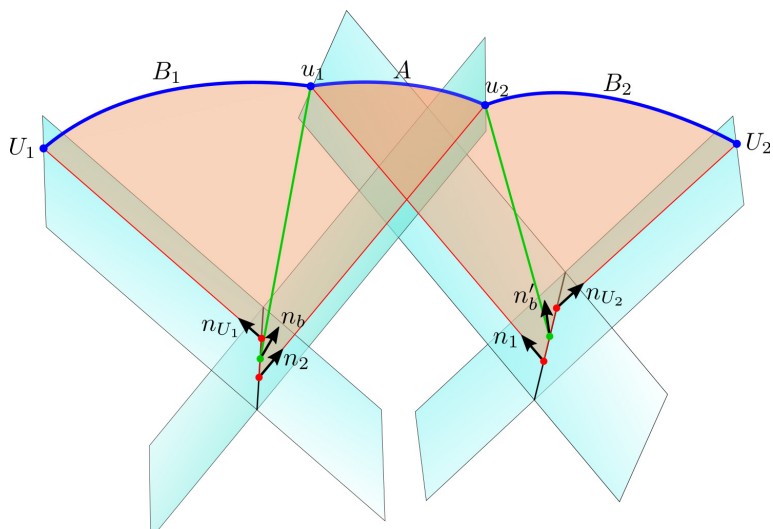

Figure 10: Construction of the EWCS for a single interval in flat-space TMG. The green line segments show only a schematic view of the actual EWCS which for the case of two adjacent intervals, require a specific regularization scheme in terms of extra null geodesics dropping from the common boundary. Note that there will be additional bulk timelike vectors on each of these extra null geodesics which come into the calculation of the EWCS.

We may now utilize the above relation to compute the correct expression of the upper bound of the EWCS for a single interval $A = [(u_1, \phi_1), (u_2, \phi_2)]$ in a thermal $GCFT_{1+1}$ defined on an infinite cylinder of circumference $\beta$ whose bulk dual is described by TMG in a FSC geometry. As shown in figure 10, the construction of the EWCS involves two auxiliary intervals $B_1 = [(U_1, \Phi_1), (u_1, \phi_1)]$ and $B_2 = [(u_2, \phi_2), (U_2, \Phi_2)]$ and correspondingly four bulk timelike vectors $n_1$, $n_2$, $n_{U_1}$ and $n_{U_2}$ directed along the null geodesics dropping from the boundary points $u_1$, $u_2$, $U_1$, and $U_2$ respectively. Now we use the adjacent intervals result in eq. (122) and take the bipartite limit of the auxilary intervals $B_1$ and $B_2$. Therefore the Chern-Simons contribution to the EWCS is given by

$$\tilde{E}_W^{\text{CS}} = \frac{1}{2\mu G_N}\left[\log\left(\frac{\beta}{\pi\tilde{\epsilon}}\sinh\frac{\pi\phi_{12}}{\beta}\right) - \frac{\pi\phi_{12}}{\beta}\right] + \frac{1}{4\mu G_N}\log 4. \tag{140}$$

Note that the constant second term in the above expression carries the signature of the Markov gap or the crossing PEE and is consistent with its geometric interpretation provided in terms of the non-trivial endpoints of the EWCS [68]. Using eqs. (30) and (140), we may now obtain the Chern-Simons contribution to the holographic entanglement negativity for the mixed state in question as

$$\mathcal{E}^{\text{CS}} = \frac{c_L}{4}\left[\log\left(\frac{\beta}{\pi\tilde{\epsilon}}\sinh\frac{\pi\phi_{12}}{\beta}\right) - \frac{\pi\phi_{12}}{\beta}\right], \tag{141}$$

where we have used eq. (84) and the constant term in eq. (140) signifying the Markov gap or the crossing PEE has been omitted. Using the Einstein gravity result in eq. (81), the complete expression for the holographic entanglement negativity becomes

$$\mathcal{E} = \frac{c_L}{4}\left[\log\left(\frac{\beta}{\pi\tilde{\epsilon}}\sinh\frac{\pi\phi_{12}}{\beta}\right) - \frac{\pi\phi_{12}}{\beta}\right] + \frac{c_M}{4}\left[\frac{\pi u_{12}}{\beta}\coth\left(\frac{\pi\phi_{12}}{\beta}\right) - \frac{\pi u_{12}}{\beta}\right], \tag{142}$$

which exactly matches with the results obtained in [56, 64]. It is interesting to note that the complete expression for the holographic entanglement negativity in the above equation can

be recast in the instructive form eq. (82), where

$$S_A = \frac{c_L}{4} \log\left( \frac{\beta}{\pi \tilde{\epsilon}} \sinh \frac{\pi \phi_{12}}{\beta} \right) + \frac{c_M}{4} \frac{\pi u_{12}}{\beta} \coth\left( \frac{\pi \phi_{12}}{\beta} \right),$$

$$S_A^{\text{th}} = \frac{c_L}{4} \frac{\pi \phi_{12}}{\beta} + \frac{c_M}{4} \frac{\pi u_{12}}{\beta}, \tag{143}$$

denote the entanglement entropy for the single interval $A$ and the thermal contribution respectively. Once again this subtraction of the thermal contribution indicates that the entanglement negativity quantifies the upper bound of distillable entanglement.

## 6 Summary and Discussion

To summarize, in this article we have advanced a novel holographic construction to obtain the extremal entanglement wedge cross section (EWCS) for several bipartite pure and mixed states in $GCFT_{1+1}$s located at the null infinity of the dual bulk (2+1)-dimensional asymptotically flat Einstein gravity and topologically massive gravity (TMG) theories. We have further proposed a prescription for the holographic entanglement negativity for the configuration in question by utilizing the EWCS obtained through the construction mentioned above. For the scenario of flat Einstein gravity, the bulk asymptotic symmetry analysis leads to the dual $GCFT_{1+1}$ with central charges $c_L = 0$, $c_M \neq 0$. In this context we have obtained the holographic entanglement negativity using our prescription through the EWCS for the configurations involving a $GCFT_{1+1}$ dual to the the bulk (2+1)-dimensional Minkowski spacetime in its ground state, a thermal $GCFT_{1+1}$ dual to the bulk non rotating flat space cosmology and a $GCFT_{1+1}$ describing a finite sized system on an infinite cylinder dual to the bulk global Minkowski orbifold. First we have obtained the EWCS for the configuration of two disjoint intervals in the dual $GCFT_{1+1}$ using our novel construction where we have utilized the length of a certain bulk extremal curve between two specific bulk geodesics. Next we proceed to the case of two adjacent intervals where the EWCS is obtained using the extremal distance between a bulk geodesic and a boundary point. This length requires a regularization as the distance of any bulk point from the boundary situated at the null infinity is still infinite. To this end, an arbitrary point on a null line descending from the boundary point is considered as a regulator. Furthermore we demonstrate that in the limit of the two disjoint intervals being adjacent we retrieve the corresponding holographic entanglement negativity for two adjacent intervals which further substantiate our holographic construction. Finally in the context of flat Einstein gravity, we have considered the pure state of a single interval in a dual $GCFT_{1+1}$ in its ground state and a $GCFT_{1+1}$ describing a finite sized system. For these scenarios the EWCS is given trivially by the length of the extremal curve homologous to the subsystem on the boundary. But for the mixed state of a single interval in a thermal $GCFT_{1+1}$ the construction is subtle. We consider the single interval in question being sandwiched between two auxiliary large intervals on both sides. Subsequently a bipartite limit described by extending the auxiliary intervals to infinity is applied by which we recover the original configuration of the single interval. A similar construction for the $AdS_3/CFT_2$ framework utilized to obtain the EWCS can be found in the existing literature. Interestingly in the case of $GCFT_{1+1}$s dual to the asymptotically flat Einstein gravity we found that the Markov gap vanishes indicating a full Markov recovery. Once again, we reiterate that a proper justification of this subtle issue requires a complete analysis of the Markov recovery process in the flat holographic setup.

Following the above computations, we extend our holographic construction to obtain the entanglement negativity for the bipartite states described earlier in a $GCFT_{1+1}$ with non-zero $c_L$ dual to the bulk flat space topologically massive gravity. The effect of non-zero $c_L$ is to

incorporate a spin for the massive particles propagating in the bulk. This results in the introduction of regulated vectors at each bulk point. Again we have obtained the holographic entanglement negativity for mixed state of disjoint intervals in the dual $GCFT_{1+1}$ by extremizing the relative orientation of such bulk vectors situated at certain bulk points. We then proceed to the case of adjacent intervals where we again consider an arbitrary point on a null line descending from the boundary point as a regulator to compute the EWCS. Taking the limit of the two disjoint intervals being adjacent we again recover the corresponding holographic entanglement negativity for two adjacent intervals. Finally for the case of the pure state of a single interval in a dual $GCFT_{1+1}$ in its ground state and a $GCFT_{1+1}$ describing a finite sized system is trivially calculated using the bulk vectors on the extremal curve homologous to the subsystem on the boundary. And for the mixed state of a single interval in a thermal $GCFT_{1+1}$ we follow the previous construction of taking a tripartite state with the single interval in question being sandwiched between two large auxiliary intervals. By taking the bipartite limit where the auxiliary intervals are extending till infinity we recover the original configuration. The expression for the holographic entanglement negativity thus obtained using our prescription matches with the corresponding replica technique results in large central charge limit. Interestingly our findings also match with the results obtained through an alternate holographic construction for the entanglement negativity involving the algebraic sum of the lengths of bulk extremal curves homologous to certain appropriate combinations of the intervals in question for both the cases, namely flat Einstein gravity and flat TMG. Remarkably, in this case we found exactly a constant Markov gap with the correct geometric interpretation in terms of the non-trivial boundaries of the EWCS. This seems somewhat counter-intuitive that only the topological part of the gravity theory in the bulk correctly captured the Markov gap or the crossing PEE for asymptotically flat spacetimes and any interpretation of this ambiguity requires a full analysis of the Markov recovery process as described in [68] or the balanced partial entanglement in [48] in such spacetimes with non-Lorentz invariant duals. This serves as a strong substantiation and consistency check for our holographic entanglement negativity prescription. We would like to emphasize here that our construction described in this work addresses the significant issue of the computation of the entanglement wedge cross section essential for the characterization of the mixed state entanglement in the context of flat space holography.

Subsequently, in appendix A we also perform a limiting analysis where we have shown that the extremal EWCS in asymptotically flat spacetimes dual to a $GCFT_{1+1}$ computed in the main text matches with the EWCS obtained through a parametric İnönü-Wigner contraction of the corresponding relativistic result obtained in the context of $AdS_3/CFT_2$ in the literature. This provides another consistency check for our holographic construction for the EWCS.

Further in appendix B, we utilize the well known fact that the flat chiral gravity is a limit of the flat space TMG for which the Newton's constant $G_N \to \infty$ such that the product of $G_N$ with the coupling constant $\mu$ of the topological term in the action is held fixed, to compute the holographic entanglement negativity for bipartite mixed state of two disjoint intervals in proximity in the dual $GCFT_{1+1}$ using our prescription. The corresponding dual $GCFT_{1+1}$ have the central charges $c_L \neq 0$, $c_M = 0$ for which the algebra is isomorphic to the chiral copy of the (relativistic) Virasoro algebra. The result thus obtained matches up to a constant Markov gap with the holographic entanglement negativity for the configuration in question reported earlier in the literature. Therefore, we find once again that a non-vanishing Markov gap or the crossing PEE originates solely from the chiral sector of the asymptotically flat gravitational theory. On a separate note, it is interesting to point out a fascinating connection of the flat space chiral gravity with the $AdS_2/CFT_1$ correspondence. A parallel could be drawn between the bulk timelike vectors introduced in the flat TMG scenario and points in the three dimensional embedding space of the Euclidean Poincare $AdS_2$. This results in the identification of the boost

required to drag the normal frame from one bulk point to another with the geodesic length between the corresponding points in the Euclidean Poincare $AdS_2$. This is an extremely interesting open avenue for future investigations as described by the progress in the corresponding $AdS_2/CFT_1$ scenario.

## Acknowledgments

We would like to thank the referee, Prof. Qiang Wen, for various fruitful suggestions and questions raised which helped in improving this manuscript.

## A  Limiting Analysis

In this appendix we will show that the extremal EWCS in asymptotically flat spacetimes dual bipartite density matrices in a $GCFT_{1+1}$ computed in the main text may be obtained through a parametric contraction of the corresponding relativistic result obtained in the context of $AdS_3/CFT_2$ in [35, 44].

The $GCA_2$ algebra can be obtained as a result of a parametric İnönü-Wigner contraction of the Virasoro algebras of a relativistic $CFT_{1+1}$:

$$t \to t, \qquad x \to \epsilon x, \tag{144}$$

with $\epsilon \to 0$. This may alternatively be written in terms of the coordinates describing the $CFT_{1+1}$ as

$$z \to t + \epsilon x, \qquad \bar{z} \to t - \epsilon x. \tag{145}$$

The central charges of the $GCA_2$ are related to those of the parent relativistic theory as

$$c_L = c + \bar{c}, \quad c_M = \epsilon(c - \bar{c}). \tag{146}$$

Now we will utilize this connection to show the equivalence of the entanglement wedge corresponding to a generic bipartite density matrix $\rho_{AB}$ in the $CFT_{1+1}$ to that in the $GCFT_{1+1}$. To this end we recall that the minimal entanglement wedge cross section for such bipartite systems in a $CFT_{1+1}$ may conveniently be expressed in terms of the cross ratios as

$$E_W = \begin{cases} \frac{c}{6} \log\left(\frac{1+\sqrt{x}}{1-\sqrt{x}}\right) & 1/2 \le x \le 1, \\ 0 & 0 \le x \le 1/2, \end{cases} \tag{147}$$

where $x = \frac{z_{12}z_{34}}{z_{13}z_{24}}$ is the cross ratio. If we allow for unequal central charges for the left and right moving sectors the above expression has the natural generalization

$$E_W = \frac{c}{12} \log\left(\frac{1+\sqrt{x}}{1-\sqrt{x}}\right) + \frac{\bar{c}}{12} \log\left(\frac{1+\sqrt{\bar{x}}}{1-\sqrt{\bar{x}}}\right). \tag{148}$$

Utilizing eq. (145), we now write the $CFT_{1+1}$ cross ratios in terms of those in the $GCFT_{1+1}$ as

$$x \to T\left(1 + \epsilon \frac{X}{T}\right), \qquad \bar{x} \to T\left(1 - \epsilon \frac{X}{T}\right). \tag{149}$$

Using eq. (149), we may now write down the fate of the entanglement wedge cross section in eq. (148) after the contraction as

$$E_W \to \frac{c}{12} \log\left(\frac{1+\sqrt{T}\left(1+\frac{\epsilon}{2}\frac{X}{T}\right)}{1-\sqrt{T}\left(1+\frac{\epsilon}{2}\frac{X}{T}\right)}\right) + \frac{\bar{c}}{12} \log\left(\frac{1+\sqrt{T}\left(1-\frac{\epsilon}{2}\frac{X}{T}\right)}{1-\sqrt{T}\left(1-\frac{\epsilon}{2}\frac{X}{T}\right)}\right). \tag{150}$$

Expanding upto linear order in $\epsilon$ and using eq. (146), the above expression reduces to

$$
\begin{aligned}
E_W^{\text{GCFT}} &= \frac{c_L}{12} \log\left(\frac{1+\sqrt{T}}{1-\sqrt{T}}\right) + \frac{c_M}{12\epsilon} \frac{\epsilon X}{2\sqrt{T}}\left(\frac{1}{1+\sqrt{T}} + \frac{1}{1-\sqrt{T}}\right) + \mathcal{O}(\epsilon) \\
&= \frac{c_L}{6} \log\left(\frac{1+\sqrt{T}}{\sqrt{1-T}}\right) + \frac{c_M}{12} \frac{X}{\sqrt{T}(1-T)} + \mathcal{O}(\epsilon).
\end{aligned}
\tag{151}
$$

Finally we use the analogues of the Brown-Henneaux formula [76] in flat holography, namely eq. (84) to obtain

$$
E_W^{\text{GCFT}} = \frac{1}{2\mu G_N} \log\left(\frac{1+\sqrt{T}}{\sqrt{1-T}}\right) + \frac{1}{4G_N} \frac{X}{\sqrt{T}(1-T)}.
\tag{152}
$$

Remarkably, this is exactly the same extremal EWCS obtained in the main text using methods of flat holography.

Next we analyze the Markov gap for two disjoint intervals in proximity in a $GCFT_{1+1}$ dual to TMG in asymptotically flat spacetime through the above parametric contraction. For the case of two disjoint intervals $A$ and $B$ in proximity in usual $AdS_3/CFT_2$ framework, we can re-express the Markov gap in terms of the EWCS and the holographic mutual information as

$$
2E_W(A:B) - I(A:B) = \frac{c}{6}\left(\log\frac{1+\sqrt{x}}{1-\sqrt{x}} - \log\frac{1}{1-x}\right) + \frac{\bar{c}}{6}\left(\log\frac{1+\sqrt{\bar{x}}}{1-\sqrt{\bar{x}}} - \log\frac{1}{1-\bar{x}}\right).
\tag{153}
$$

Now performing the İnönü-Wigner contraction of the cross ratios as given in eq. (149), we obtain

$$
2E_W(A:B) - I(A:B) = \frac{c_L}{6}\left(\log\frac{1+\sqrt{T}}{1-\sqrt{T}} - \log\frac{1}{1-T}\right) + \frac{c_M}{6}\left(\frac{X}{\sqrt{T}(1-T)} - \frac{X}{1-T}\right),
\tag{154}
$$

where we have used eq. (146). We can clearly see from the above expression that as $T \to 1$, the difference becomes proportional to $\log 2$ coming from the first parenthesis. We emphasize that although a geometric interpretation in terms of the number of non-trivial boundaries of the EWCS is still elusive for the case of Einstein gravity in the bulk, the above limiting analysis consistently shows that the Markov gap should vanish in such scenarios. Note that the Markov gap or the crossing PEE originating from the purely topological part of the action is still elusive and a complete understanding of this behaviour requires a proper analysis of the Markov gap or the BPE in the flat holographic setup.

# B EWCS in Flat Chiral Gravity: Connection to $AdS_2/CFT_1$

In this appendix we deal with a special case of flat space TMG, called the flat space chiral gravity (F$\chi$G) or sometimes conformal Chern-Simons gravity [77–79] with the non-covariant action:

$$\mathcal{S}_{CSG} = \frac{k}{4\pi} \int d^3x \, \sqrt{-g} \left[ \varepsilon^{\alpha\beta\gamma} \Gamma^\rho_{\alpha\sigma} \left( \partial_\beta \Gamma^\sigma_{\gamma\rho} + \frac{2}{3} \Gamma^\sigma_{\beta\eta} \Gamma^\eta_{\gamma\rho} \right) \right], \tag{155}$$

where $k$ is the Chern-Simons level. The F$\chi$G may be obtained as a limit of the flat space TMG action in eq. (83) by scaling the Newton's constant to infinity, $G_N \to \infty$, while keeping $\mu G_N \equiv 1/8k$ fixed. The equation of motion for F$\chi$G leads to a vanishing Cotton tensor, thereby confirming that the solutions admitted by the action (155) are conformally flat. It was demonstrated in [77] that the dual quantum field theory is described by a $GCFT_{1+1}$ with central charges $c_L = 24k$, $c_M = 0$ whose algebra is isomorphic to the chiral copy of the usual Virasoro algebra.

Next we will describe a construction for computing the extremal EWCS in F$\chi$G. The Chern-Simons action (155) describes a massless spinning particle propagating in the bulk three dimensional spacetime, for which the on-shell worldline action in given in eq. (86), where the symbols inherit their usual significance from the TMG case. The only degrees of freedom are given by the timelike vectors $n_i$ erected at each bulk point $y_i$. The construction of the entanglement wedge corresponding to two generic (disjoint) intervals $A$ and $B$ on the asymptotic boundary is identical to that described in subsection 5.1, except that now the cross section only picks up a topological contribution. The cross section is obtained by choosing two arbitrary bulk points $y_b$ and $y_b'$ on the extremal curves computing the entanglement entropy of $A \cup B$ and then extremizing the Galilean boost required to drag the the bulk vector $n_b$ at $y_b$ to $y_b'$:

$$E_W^{F\chi G} = 2k \min_{n_b, n_b'} \text{ext} \, \Delta\eta(n_b, n_b'). \tag{156}$$

Therefore, for two disjoint intervals in proximity in the ground state of the $GCFT_{1+1}$ dual to F$\chi$G in Minkowski spacetime, the extremal EWCS is obtained as

$$E_W^{F\chi G} = 2k \log\left( \frac{t_{13} t_{24}}{t_{14} t_{23}} \right) + 2k \log 4, \tag{157}$$

where we have used eq. (96). We may now utilize a version of our flat-holographic proposal in eq. (30) restricted to the case of F$\chi$G to obtain the corresponding entanglement negativity as

$$\mathcal{E} = \frac{c_L}{8} \log\left( \frac{t_{13} t_{24}}{t_{14} t_{23}} \right), \tag{158}$$

which matches exactly with that reported in [64] providing a consistency check. Note that, as earlier we have subtracted the constant Markov gap (or crossing PEE) from the EWCS while computing the entanglement negativity. We emphasize that the above analysis substantiates the fact that the constant Markov gap originates solely from the chiral sector of the gravity theory.

It is now straightforward to extend this construction for the entanglement wedge for the cases of two adjacent intervals and a single interval in the boundary theory. Instead, we would like to report a fascinating connection of F$\chi$G with the $AdS_2/CFT_1$ correspondence. As noted in [61], the bulk timelike vectors of the form (90) can be thought of as points in the three dimensional embedding space of Euclidean Poincare $AdS_2$, with $x$ and $u$ being the corresponding Poincare coordinates. With such a connection, the boost in eq. (86) may be identified with the geodesic distance in Eucildean Poincare $AdS_2$, between the points corresponding to $n_i$ and

$n_f$. Therefore the extremal EWCS in F$\chi$G may also be computed using the worldline method in Euclidean $AdS_2$. This observation also finds support in the fact that the dual $CFT_1$ accomodates only one copy of the Virasoro algebra, which is isomorphic to the $GCA_2$ for $c_M = 0$. It was also demonstrated in [77] that the representations of such $GCA_2$ reduce to the Virasoro module, as well as the possibility of putting unitarity constraints. Therefore, one might wonder about a holographic correspondence between a topological gravitational theory with a specific unitary field theory in two dimensions lower.

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
