# Peer review of "Entanglement Wedge in Flat Holography and Entanglement Negativity"

_SciPost Physics Core, doi:SciPost Phys. Core 5, 013 (2022)_

## Round 1 · Referee Report · Qiang Wen · 2021-9-21

Strengths

1, discussed new aspects of the 3d flat holography

2, results can be compared with the results from other independent methods

3, a non-trivial support for the novel geometric picture for entanglement and correlation functions in flat holography, which looks quit different from the RT formula.

Weaknesses

1, there are several subtle points the authors need to clarify before I can fully trust their proposal.

2, the summary and discussion section just repeated what the authors have done.

Report

The 3d flat holography is the correspondence between the so-called Galilean CFT$_2$ and 3-dimensional asymptotic flat spacetime with asymptotic symmetries captured by the BMS$_3$ group. Based on the novel geometric picture for entanglement entropy [60] and correlation functions [61] in flat holography, and the proposal of the duality between holographic entanglement negativity and the area of the EWCS (times 3/2) in the context of AdS/CFT [35,36], the authors propose that the correspondence between the negativity and EWCS can be extend to flat holography. They calculated the EWCS in various configurations and find consistency with the negativity calculated by replica trick in GCFT.

This work makes an important contribution to the study of holography beyond AdS/CFT. The novel geometric picture for holographic entanglement entropy in non-AdS holography indicates that the mathematic structure behind flat holography is quite different from the AdS case. This paper proposed a natural way to define the entanglement wedge and EWCS in flat holography based on the novel geometric picture. The results look reasonable and is reported to be consistent with the results from other independent methods. On other hand, this work gives a non-trivial test for the novel geometric picture in non-AdS holography. However, several subtle points (see Requested changes) in the paper need to be clarified before I can recommend publication.

Requested changes

1. The novel geometric picture for entanglement and correlation functions is not familiar to most of the people in the quantum gravity community. Since it plays an extremely important role in this work, I suggest the authors to give it a more explicit background introduction.

2. The null geodesics descending from the endpoints of the boundary interval are not unique. The author should clarify how to specify those null geodesics.

3. The authors reported that, the EWCS exactly matches with the linear combinations of the lengths of extremal curves, which was previously conjectured [62] to be the holographic dual of negativity by some of the authors. For example in the adjacent interval cases, this linear combination is just proportional to the mutual information. However, in the AdS case this matching is not exact. The difference between the mutual information (times 3/4) and the EWCS (times 3/2) is a constant $c/3\ln 2$, which will not vanish when taking the large c limit. It is called the crossing PEE in 2103.00415, the Markov gap in 2107.00009, and also studied in the condensed matter community in 2011.11864. However, in the flat case the authors claim that this constant difference disappears. I would like to suggest the authors to give a brief interpretation for that.

4. I donot quite understand the adjacent limit of the disjoint configuration. No matter how $x_2$ approaches $x_3$, the extremal surface $\gamma_{23}$ should be determined. However, in Fig. 5 the coordinate $y_b$, which is supposed to be settled on $\gamma_{23}$, is a free parameter along the null geodesic descending from the endpoint $x_2$. Also (44) is sensitive to how we take the limit $x_{23}\to 0$ and $t_{23} \to 0$. I think the authors should clarify how they explicitly take the limit for (44) to get (64)?

5. Another point I concern is the limiting analysis in Appendix A. It was shown in [60] that when taking the flat limit in the AdS space, the outer horizon is pushed infinitely far away, hence all the geometric quantities (including the EWCS) we discuss in flat holography should be the flat limit of geometric quantities inside the BTZ outer horizon (see section 7 in [60]). However in the Appendix A, the limit was taken directly on the AdS results. If the EWCS is just a limit of the EWCS in the AdS case, why the matching between EWCS and negativity is exact in the flat case, while not the AdS case?

Also I list the typos I found

1, the second line below Fig.5 should be ``the distance from a bulk point to the boundary ...''

2, the two terms in the right hand side of the second inequality in (77) shoule be $\frac{1}{2}I(A,B_1)+\frac{1}{2}I(A,B_2)$

  • validity: ok
  • significance: high
  • originality: good
  • clarity: ok
  • formatting: good
  • grammar: good

Author:  Gautam Sengupta  on 2021-11-03  [id 1905]

(in reply to Report 1 by Qiang Wen on 2021-09-21)

We would like to thank the referee for the interesting questions and comments. Our detailed response to the issues raised and their resolutions are described below. We have appropriately made the required changes to our manuscript as suggested by the referee.

1) We have appropriately modified and expanded the introduction in our revised manuscript to provide further background details about the geometric construction for entanglement in flat holographic scenarios. Further the discussion about the geometric construction of the extremal curves in subsection 2.2 following eq. (15) has been elaborated in the revised manuscript.

2) We thank the referee for raising this subtle issue about the ambiguity in determining the null geodesics descending from the endpoints of the interval at the asymptotic boundary. As noted in reference [63] of our revised manuscript, this non-uniqueness of the null geodesics subsequently translates to an ambiguity in the value of the entanglement entropy and essentially corresponds to a choice of a cut-off surface at the asymptotic infinity. Hence although the entanglement entropy is finite, it really involves a cut-off dependence through different choices of the null geodesics which may be accounted for by introducing suitable bulk translations. We adopt the same approach in our analysis and choose the cut-off surface such that the length of the extremal curve homologous to an interval $A$ at the boundary is given by eq. (14) in our revised manuscript with any inconsistencies being accounted for through such bulk translations mentioned above. We have also added a brief discussion in section 2.2 following eq. (15) of the revised manuscript where we address this ambiguity in the choice of the null geodesics as suggested by the referee.

3) We would like to thank the referee for highlighting this crucial issue regarding the difference between the EWCS and half the mutual information in the context of the usual AdS/CFT scenario which is termed as the crossing PEE or the Markov gap. We had, in fact, also encountered this constant difference between the EWCS and the entanglement negativity (which as the referee correctly pointed out, is proportional to the mutual information in the adjacent case). However, at the time of the submission of this article to the arXiv this issue was not established in sufficient details and hence it escaped our notice. Consequently, we had neglected the difference as an insignificant constant.

In accordance with the referee's suggestion we have examined this issue in details and we have found an interesting behaviour of this Markov gap (crossing PEE) in our work. For the case of bulk purely Einstein gravity in asymptotically flat spacetimes, the Markov gap vanishes indicating a full Markov recovery, while it remains non-zero in the case of topologically massive gravity (TMG). However the appearance of the Markov gap is consistent with the geometric description in reference [68] of our revised manuscript, extended to the case of flat space holography (See also our response to the fifth point below). We have discussed the modifications to the holographic proposal for the entanglement negativity due to the appearance of Markov gaps following eq. (29) and eq. (30) of our revised manuscript. Furthermore, we have addressed these issues in the summary and discussion section of our revised manuscript. We have also explicitly pointed out the appearance (or absence) of the Markov gap in each of the cases discussed and added a limiting analysis of the same in appendix A of our revised submission.

We would like to point out that a complete analysis of the EWCS including the Markov gap in the case of a single interval at finite temperature for TMG in asymptotically flat spacetimes requires further investigation. In the present work, we only obtained an upper bound to the EWCS for the above configuration. We have properly modified the corresponding analysis in subsection 5.3.3 and found that once again, apart from a constant, the resulting EWCS (times 3/2) matches with the holographic entanglement negativity obtained in reference [56,64] of our revised manuscript.

4) The referee has correctly pointed out that irrespective of the exact process for the adjacent limit of $x_2$ approaching $x_3$, the extremal surface $\gamma_{23}$ is uniquely specified and the point $y_b'$ in Fig. 4 lies on this extremal surface. In the limit $x_2 \to x_3$, the extremal surface $\gamma_{23}$ shrinks to a single point $\gamma_{23} \equiv y_b^\textrm{extr}$. However we would like to emphasize that upon extremization over the free parameter labelled $y_b$ in Fig. 5 it is precisely identical to the point $y_b^\textrm{extr}$. Hence the extremization procedure described in our submission consistently leads to an identical construction to the one arrived at from a limiting analysis of the corresponding disjoint configuration. We have included a discussion regarding this subtle issue in subsection 4.2 in our revised submission.

With regard to the second part of the referee's comment we emphasize that in order to have a finite entanglement negativity, one should actually subtract cut-off dependent divergent terms of $\mathcal{O} (\frac{\epsilon_x} {\epsilon_t})$ from each of the terms of the form $\frac{x}{t}$ which appear in the expressions for the entropies as described in reference [60] of our revised manuscript. In our article we chose to omit such divergent terms from the general expressions for brevity and consistently disregarded the $\mathcal{O}(\frac{\epsilon_x}{\epsilon_t})$ term arising from the quantity $\frac{x_{23}}{t_{23}}$ in eq. (44) for the adjacent limit. This also extends to the other cases described in our work. As suggested by the referee we have included a footnote (footnote 7 on page 21) in the revised manuscript, briefly discussing this issue for the process of the adjacent limit of the disjoint intervals result .

5) We agree with the referee that the most general phase space of $AdS_3$ solutions with Brown-Henneaux boundary conditions is given by the BTZ black hole and consequently the flat space limit should be taken by pushing the outer horizon to infinity. However, we would like to point out that in our work, the limiting analysis has been performed on the geometric quantities in the bulk re-expressed in terms of the field theory data, in particular the cross-ratios involved. Such parametric contraction of the cross ratios, as expressed in eq. (149) of our revised manuscript, does not require the explicit bulk description. Rather, as also pointed out in reference [44] in the case of usual $AdS_3/CFT_2$ scenario, one may obtain for example, the EWCS by performing a conformal transformation to obtain the modified cross-ratios.

We would like to clarify that although the limiting analysis to obtain the EWCS from the corresponding relativistic results is perfectly valid, we had previously disregarded an analysis of the Markov gap or the crossing PEE. In our revised manuscript, we have attempted to perform a limiting analysis of the Markov gap in appendix A and found that it originates exclusively from the topological part of the action. We emphasize that although a geometric interpretation in terms of the number of non-trivial boundaries of the EWCS is still elusive for the case of Einstein gravity in the bulk, the limiting analysis consistently shows that the Markov gap should vanish in such scenarios.

6) The typographical errors have been appropriately corrected.

---

## Round 2 · Referee Report · Qiang Wen (Referee 1) · 2021-11-13

Report

I thank the authors for their revision and I am satisfied by their replies on the comments 1,3,4,5.

Especially, their reply on comment 3 gives interesting relation between the entanglement negativity (or the EWCS, reflected entropy) and the mutual information. This shows that, unlike the case in AdS/CFT, the so called Markov gap or crossing PEE vanishes in BMSFTs dual to the Einstein gravity (where the central charge $c_{L}=0$). This broadens the interestingness of the manuscript.

My only concern is still the comment 2. Originally, I just want the authors to specify that, those null rays in the geometric picture are the lines emanating from the boundary endpoints and moving along the bulk modular flow. In the Bondi gauge they are just lines along the r coordinate. However the authors introduced the "ambiguity" raised up by Ref. [63]. I cannot fully understand this “ambiguity”, because after a translation along the 'y' coordinate the image of the boundary interval no longer lines on the boundary at a constant 'r'. I kindly suggest the author to remove this part as it is controversial.

The manuscript is significantly improved and I am happy to recommend it for publication.
  • validity: -
  • significance: -
  • originality: -
  • clarity: -
  • formatting: -
  • grammar: -

Author:  Gautam Sengupta  on 2022-01-12  [id 2093]

(in reply to Report 1 by Qiang Wen on 2021-11-13)

We would like to thank Referee1 for the comments and for recommending publication. As per the Referee's suggestion we have removed the part described as controversial by the Referee in the resubmitted manuscript.

---

## Round 2 · Referee Report · Anonymous (Referee 2) · 2022-1-3

Report

I am the second referee; the paper has already gone through one round of review and revision.

I was asked by the editor to address a question concerning the computation of the negativity using twist operators. This computation was done in a different paper (ref 64 in the current draft). In the current paper the corresponding EWCSs were computed and compared to the results of that computation, finding agreement (up to some constant terms in the TMG case). While it is interesting to ask about the twist operator computations and their validity, in my view this question is not directly germaine to the issue of whether this paper ought to be published. Even if the twist operator computations turn out to be flawed, the EWCS computations of the current paper will still be valuable (perhaps even more so, since it could indicate a disagreement between the EWCS and negativity). Therefore I don’t think that this issue should prevent the current paper from being published.

I agree with the comments of the previous referee, and I also agree with their conclusion that the paper deserves to be published in SciPost.
  • validity: -
  • significance: -
  • originality: -
  • clarity: -
  • formatting: -
  • grammar: -

Author:  Gautam Sengupta  on 2022-01-12  [id 2092]

(in reply to Report 2 on 2022-01-03)

We would like to thank Referee2 for the comments and for recommending publication.

---

## Round 2 · Author Response

We have examined the referee report for our submission scipost_202107_00038v1 entitled ``Entanglement Wedge in Flat Holography and Entanglement Negativity". We would like to thank the referee for the interesting questions and comments. Our detailed response to the issues raised and their resolutions are described in the reply to the referee's report. We have appropriately made the required changes to our manuscript as suggested by the referee.

---

## Round 2 · List of Changes

We have provided the complete list of changes made in the authors response section.

---

## Round 3 · Author Response

We have made the minor modification sought by the Editor and removed the part from the paragraph after eqn.(15)
on Page 8, as suggested by the Referee1.

---

## Round 3 · List of Changes

Please see the Author Comments.

---

## Editorial Decision

published